# Modeling the electrical resistivity of polymer composites with segregated structures

Sung-Hoon Park[1,8], Jinyoung Hwang [2,8], Gyeong-Su Park[3], Ji-Hwan Ha[1], Minsu Zhang[4], Dongearn Kim[5], Dong-Jin Yun[6], Sangeui Lee[7] & Sang Hyun Lee[4]

Hybrid carbon nanotube composites with two different types of fillers have attracted considerable attention for various advantages. The incorporation of micro-scale secondary fillers creates an excluded volume that leads to the increase in the electrical conductivity. By contrast, nano-scale secondary fillers shows a conflicting behavior of the decreased electrical conductivity with micro-scale secondary fillers. Although several attempts have been made in theoretical modeling of secondary-filler composites, the knowledge about how the electrical conductivity depends on the dimension of secondary fillers was not fully understood. This work aims at comprehensive understanding of the size effect of secondary particulate fillers on the electrical conductivity, via the combination of Voronoi geometry induced from Swiss cheese models and the underlying percolation theory. This indicates a transition in the impact of the excluded volume, i.e., the adjustment of the electrical conductivity was measured in cooperation with loading of second fillers with different sizes.

[1] Department of Mechanical Engineering, Soongsil University, Seoul 06978, South Korea. [2] School of Electronics and Information Engineering, Korea Aerospace University, Goyang-si 10540, South Korea. [3] Department of Material Science and Engineering and Research Institute of Advanced Materials, Seoul National University, Seoul 08826, South Korea. [4] School of Electrical Engineering, Korea University, Seoul 02841, South Korea. [5] Korean Institute of Industrial Technology, Incheon 21999, South Korea. [6] Samsung Advanced Institute of Technology, Samsung Electronics, Suwon 16678, South Korea. [7] Department of Mechanical Engineering, Inha University, Incheon 22212, South Korea. [8]These authors contributed equally: Sung-Hoon Park, Jinyoung Hwang. Correspondence and requests for materials should be addressed to S.-H.P. (email: leopark@ssu.ac.kr) or to S.H.L. (email: sanghyunlee@korea.ac.kr)

There has been a great interest in the study of conducting polymer composites in various applications, such as electromagnetic interference shielding, electronic packaging, structural reinforcement, radar absorption, heating elements, and high-charge storage capacitors, etc.[1–5]. High aspect-ratio fillers that attain the electrical percolation at low-volume fractions are exceedingly desirable for their improved structural integrity and electrical conductivity of the resulting composites[6,7]. Carbon nanotube (CNT)-based polymer composites have been extensively investigated in that CNTs offer with very attractive options, in particular, for their extremely large aspect ratio and expanded surface area[8]. For practical usage of CNT composites, chemical functionalization approaches (between functional groups on CNT and polymer) and effective mixing techniques, such as three-roll milling, were suggested to achieve uniform dispersion of CNTs in the polymer matrix[7,9,10].

Hybrid CNT composites comprised of multiple types of filler have recently received considerable attention for their ability to improve electrical and mechanical properties[11–14]. The incorporation of micro-scale second fillers creates the excluded volume that effectively leads to a segregated network of nanotubes[15,16]. In consequence, the probability of CNTs forming a conducting network increases in the segregated composite system, indicating that much lower CNT loading suffices to reach a certain degree of the electrical conductivity. Thus, as compared to CNT composites made only of CNT fillers, the segregated composite system could effectively relieve the cost issue of CNTs. In a mechanical point of view, dramatic strengthening and toughening, which are essential requisites for practical applications, can be achieved by the incorporation of particulate fillers, such as silica, $CaCo_3$, and $BaSO_4$[17].

However, the effect of the addition of secondary fillers is still under controversy due to its conflicting behaviors. While enhanced electrical conductivities are attained using secondary filler systems based on the excluded volume theory[11,12,15,16,18,19], the decreased electrical conductivities are also observed because of the prevention of the interconnection between CNTs despite their similarity in structure and materials[20–22]. Although there have been successful trials with theoretical models in second-filler composite systems[15,23–29], these two opposite tendencies

((1) increased conductivity due to the reduction in free volume of the polymer matrix and (2) decreased conductivity due to the disruption of CNT networks) are not well understood. In addition, it remains unanswered how the dimension of second fillers affect the electrical conductivity of the second-filler composite systems.

This work aims at comprehensively understanding the effect of the dimensional properties of secondary particulate fillers on the electrical conductivity of CNT polymer composites. Conventional approaches with the excluded volume concept over fiber random networks, which only capture the monotonicity of the composite percolation, have a limitation in explaining the behavior transitions in electric conductivity of the composite systems. To this end, Swiss cheese model, which enables to account for the distribution and dimension of secondary fillers, is introduced to overcome the failure in the description of the composite system behaviors by the excluded volume. The opposite electrical behaviors from the addition of secondary fillers can be described through the combination of Swiss cheese model and fiber random networks as shown in Fig. 1. From a conducting fiber composite having secondary particulate fillers (Fig. 1a), the distribution of secondary particulate fillers on polymer matrix (Fig. 1b), which corresponds to the shape of Swiss cheese with many pores (Fig. 1c), is separated. The void representing to a pore is associated with a single secondary filler where conducting fibers are excluded, and the volume of "cheese" is the space that conducting fibers occupy. In this Swiss cheese model, a Voronoi tessellation associated with each of voids is the region consisting of all points closer to the center of the void than to those of any other voids (Fig. 1c). A network formed with the collected sides of Voronoi tessellations (shown in gray in Fig. 1c) represents the collection of all conducting paths in the polymer matrix with particulate fillers. The CNT network is separated from the conducting fiber composite in Fig. 1a to evaluate the effective conductance of conducting paths formed with the edges of Voronoi tessellations (Fig. 1d). The local conductance of a point within the volume of Swiss cheese model, which depends on the density of CNT fibers placed around the corresponding region is represented according to the brightness of the color (Fig. 1e). The darker the red color of a point is, the denser the CNT fibers are located, indicating higher

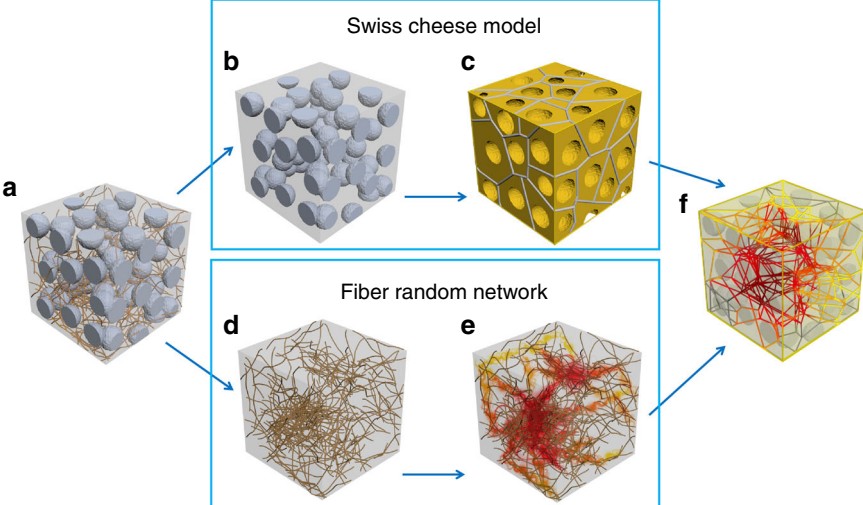

**Fig. 1** Schemes of the size effect of secondary particulate fillers on the electrical conductivity of fiber/polymer composites. **a** A conducting fiber composite having secondary particulate. **b** The distribution of spherical secondary fillers on polymer matrix. **c** The corresponding Swiss cheese having pores (spherical secondary fillers) with Voronoi geometry (having pipe line network that all possible configuration of conducting path). **d** The distribution of conducting fibers obtained by removing secondary fillers. **e** The degree and formation of electrical network by local conductance. **f** The combination of the fiber random network and Voronoi tessellation that corresponds to effecting conducting network

local conductance. To observe the degree and formation of an electrical network and to determine the total conductance of the system, an available configuration of the conducting path network in Fig. 1c and the density shown in Fig. 1e are combined by masking the density over the path network. The collection of the resulting conducting paths in darker red color acts as an effective conducting network. Thus, the total conductance can be evaluated by considering the continuum percolation property of the effective network.

In this study, the transition in the impact on the electrical conductivity is observed in accordance with the excluded volume caused by secondary particles, i.e., the excluded volume improves/degrades the conductivity of CNT composites depending on the particle size. This sets up a hypothesis that the network morphology of CNT composites depends on the size of the second fillers, leading to the variation of the electrical conductivity. The hypothesis is proved in experimental and theoretical ways. This unifying framework enables an integrated approach of handling two conflicting conducting behaviors of polymer composites, which have been addressed separately.

## Results

**Numerical characterization of polymer composites.** The Monte Carlo (MC) simulation is carried out for the prediction of percolating behaviors of the CNT network in the CNT/silica composites under various size and concentration conditions of composite components, and the electrical conductivity of the resultant network is also derived. Note that, in this simulation, a flexible CNT wire in the composites is modeled as a long thin cylindrical object with many joints in order to consider the bending of the wire.

Spherical volumes representing silica particles are scattered in a cubic domain by choosing at random central coordinates of individual particles apart from each other farther than the diameter. The diameter of the sphere is determined straightforwardly from the average size of the silica filler, and the number of spheres per unit volume is obtained from the weight ratio of the silica in the composite samples used in the experiment. Upon the completion of the random placement of spherical fillers, a CNT network is created on the volume by placing multiple random instances of a flexible CNT wire. The flexibility is reflected in the wire by modeling it as a series of small line segments contiguously connected with joints that are allowed to bend freely within a certain degree. To create a single flexible wire on the simulation domain, constituting line segments are placed in sequence from one end to the other. The first line segment is placed at random in the volume that spherical fillers do not occupy. Subsequent line segments keep being placed in a straight line unless they touch pre-existing spheres. If the candidate position of the following line segment is inside a sphere, the direction of the line segment is bent toward a tangential direction of the sphere in contact so that the flexible wire smoothly circumvents the sphere. In this simulation, the width and length of a single wire are obtained from the average diameter and length of CNTs used in the experiment, respectively. The length of a line segment contained in a flexible wire is set identical to the width of the wire so that the contact of two flexible wires can be examined simply (see Supplementary Figs. 1 and 2). The rotation range of the joint between two consecutive line segments is limited to 120° based on the physical property of a CNT so that a perfect hexagonal structure of a CNT can be preserved without atomic destruction[30–33]. The overall procedure is summarized in Supplementary Fig. 1.

Upon the construction of a random representation of the CNT/silica composites, the connection configuration of the CNT network is explored by examining contacts between wires. The connectivity between two wires is determined by calculating all possible pairwise

distances between line segments belonging to respective wires. A graphical representation of the procedure is presented in Supplementary Fig. 2. Based on the connectivity information of the CNT network, a clustering analysis is carried out to identify percolating clusters of wires that span across the simulation domain. Starting from the leftmost wires touching the left side of the cubic simulation domain, each cluster is expanded gradually toward the right side by including wires in contact. If a cluster contains one of the rightmost wires touching the right side of the simulation domain, this cluster acts as a conducting path, i.e., a percolating cluster. By contrast, wires belonging to non-percolating clusters are excluded from the network resistance calculation.

Finally, the overall resistance of the network is calculated by applying Kirchhoff's current law (KCL) at all junctions where wires contact each other[34]. To be specific, two unknowns are introduced to represent the voltages at two points that form a contact, which is replaced by a resistor. A linear equation is established via KCL by taking account of the voltage drop across the contact due to the contact resistance between CNTs[35]. If KCL is applied at each unknown, as many linear equations as the number of unknowns associated with all contacts contained in the cluster are obtained. Thus, the resulting system of linear equations can be always solved by back substitution. The solution of the system of linear equations provides the total current flow into the domain under an applied voltage of 1 V across the simulation domain, and the inverse of the results corresponds to the overall resistance of the network.

**Morphological analysis of polymer composites.** Ideal dispersion of two fillers (CNT and silica) in polymer matrix is critical in matching theoretical computational modeling. In particular, poor dispersion conditions of two fillers often cause opposite or random trends in electrical conductivity despite their similar structure and materials. In order to obtain consistent composite conductivity for given CNT and silica wt%, both fillers should be dispersed uniformly within the polymer matrix. Uniform dispersion was achieved through the use of three-roll milling process, which can induce high shearing forces to highly entangled nanoparticles[9,10]. Three rolls rotating in opposite directions with different speeds disentangle bundles of CNTs and nano/micro-size silica particles. This process results in the dispersion conditions of the fillers, which can be observed from scanning electron microscopy (SEM) images illustrated in Fig. 2a for the CNT composite without silica, Fig. 2b for the CNT composite with nano-size fillers (silica 30 wt%), and Fig. 2c for the CNT composite with micro-size fillers (silica 40 wt%), respectively. Due to the limitation of the mixing process regarding viscosity and brittleness issues, the maximum loading contents of nano-silica are limited to 30 wt%, while 1 wt% CNT loading was applied. Although uniform dispersion conditions of CNTs and silica particles were observed in SEM images, transmission electron microscopy (TEM) analysis was conducted to examine the configuration of CNTs and silica particles in polymer matrix. TEM images of composites and their graphical schemes are shown together in Fig. 3. It is observed in the TEM image of Fig. 3b that, in CNT composites without silica particles, CNTs are of a relatively linear shape in the polymer matrix, corresponding to Fig. 3a. On the other hand, the TEM image of Fig. 3d shows that CNTs in the nano-silica mixture are mostly twisted, corresponding to Fig. 3c. Figure 3e depicts micro-silica particles in the composites leading to the concentration of CNTs as observed in Fig. 3f. Morphological analysis reveals the shortening of CNTs after three-roll milling. As-received CNTs are 10–15 μm long as shown in Supplementary Figs. 3 and 4. However, the CNT length measured after three-roll milling reduces to the range of 3–6 μm.

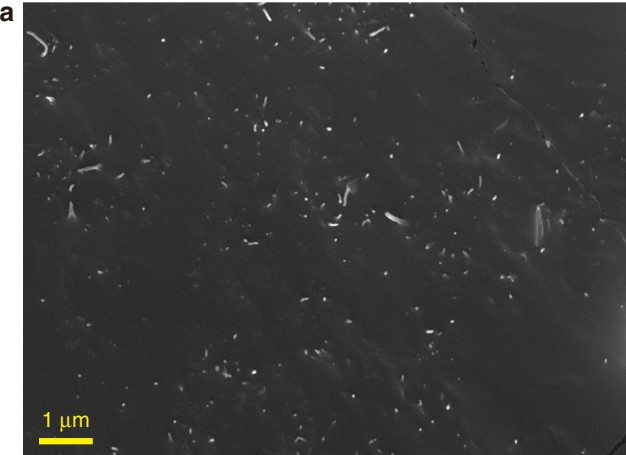

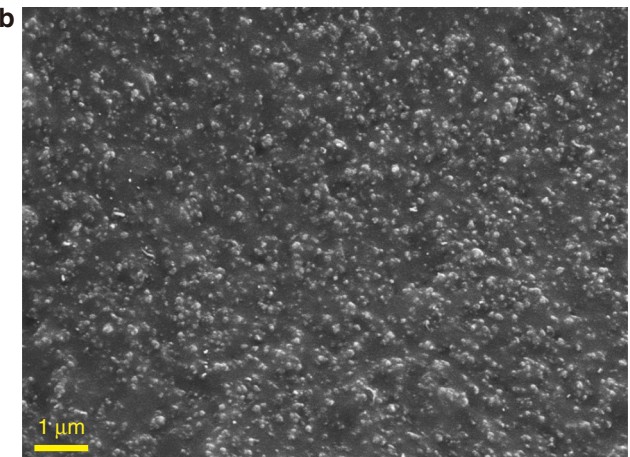

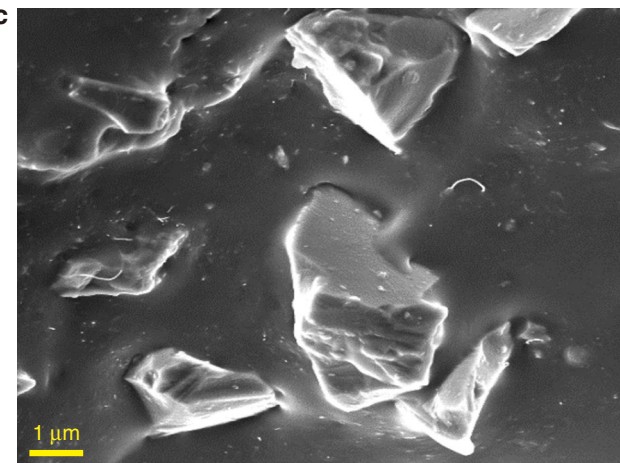

**Fig. 2** SEM images of CNT composites. **a** CNT composite without silica fillers. **b** Nano-silica fillers with diameter of 20 nm. **c** Micro-silica fillers with diameter of 3–4 μm

According to Supplementary Fig. 3, the length of CNT is specified to 5 μm for computational modeling. Along with CNTs, there is a decrease in the micro-silica particle size due to the milling process. The size of as-received micro-silica particles ranges between 7 and 10 μm as shown in Supplementary Figs. 5 and 6. The milling process causes the reduction of the micro-silica size to 2–4 μm (mostly measured around 3 μm). From the distribution evaluated as in Supplementary Fig. 5, the size of micro-silica is considered to be 3 μm for simulation.

**Effect of secondary fillers on electrical conductivity**. To investigate the size effect of secondary particulate fillers on the electrical resistivity of the composites, the CNT content is fixed to 1 wt%, which correspond to above the electrical percolation[7,10]. The electrical conductivity is monitored with varying silica (as a secondary particulate filler) contents. Figure 4a shows that, in the case of the CNT composites containing micro-size silica fillers, increasing micro-size silica filler contents decreases the resistivity of the composite. Previous studies explain that the improvement in conductivity is attributed to the excluded volume effect of particulate fillers, which leads to a concentrated CNT network[11,12,15,19]. By contrast, in the case of the CNT composites containing nano-size silica fillers, the resistivity increases as the silica filler contents increase, indicating that the excluded volume effect does not work. This opposite effect of the exclude volume is often observed from previous literature implying that current states of the art fail to explain these conflicting behaviors[20–22]. The increased resistivity is attributed to the disruption of the CNT network such as twists of CNT caused by dispersed nano-size silica particles as in Fig. 3c, d. Raman spectra illustrated in Supplementary Fig. 7 also confirm the effect of structural changes and the defect of CNTs on the resistivity of the composites. The invariance in the D-band and G-band peaks for all tested samples reveals that no atomic destruction occurs in CNTs and the resistivity change is not attributed to such atomic structure change.

To observe the degree of the difference, CNT contents are increased under the fixed secondary silica contents (for micro-silica 40 wt% and nano-silica 30 wt%). The resistivity changes of the composite are presented in Fig. 4b. The particulate filler effect on the resistivity of the composite is maximized in low CNT loading contents, while particulate fillers do not show a significant impact in high CNT loading contents. All three-type composites follow a power-law characteristic ($\sigma_{DC} \sim \sigma_0 (p - p_c)^{\beta}$) of the electrical percolation behavior, where $p_c$ is the percolation threshold, $p$ is the volume fraction of the conducting filler, $\sigma_{DC}$ is the conductivity of the composite, $\sigma_0$ is a constant parameter, and $\beta$ is a critical exponent[10,36–40]. From the best fit of the $\sigma_{DC}$ data to log–log plots of the power law, the value of $\beta$ is evaluated (the nano-silica composites of $\beta \sim 2.7$, the CNT composites without silica of $\beta \sim 1.6$ and the micro-silica composites of $\beta \sim 1.2$ are obtained, as shown in Supplementary Fig. 8).

## Discussion

The experimental results of the resistivity are further investigated by computational modeling (the MC simulation), by which the electrical resistivity variation according to the network topology can be predicted. Random instances of the CNT/silica composites generated from the simulation are presented in Fig. 5. Figure 5a shows a random network of CNTs with the diameter of 15 nm and the length of 5 μm in the simulation domain of 10 μm × 10 μm × 10 μm. In the simulation, the CNT content of 1 wt% dispersed in the epoxy resin leads to the resistance of $3.6 \times 10^4$ Ω, which is consistent with the resistivity value of 0.36 Ω m. Figure 5b, c illustrates representative random instances of the CNT/silica composites generated with the same loading amount of CNTs as in Fig. 5a along with the micro-size (the diameter of 3 μm) and the nano-size (the diameter of 50 nm) particulate fillers, respectively, at the silica content of 30 wt%. In comparison with the CNT network without silica fillers, the CNT network formed in the micro-silica composite yields several dense CNT clusters owing to the excluded volume arising from the micro-size silica fillers, which is consistent with the excluded volume theory addressed in previous studies. The topological change of the network leads to the enhancement in the probability of CNTs being interconnected. The total number of interconnecting

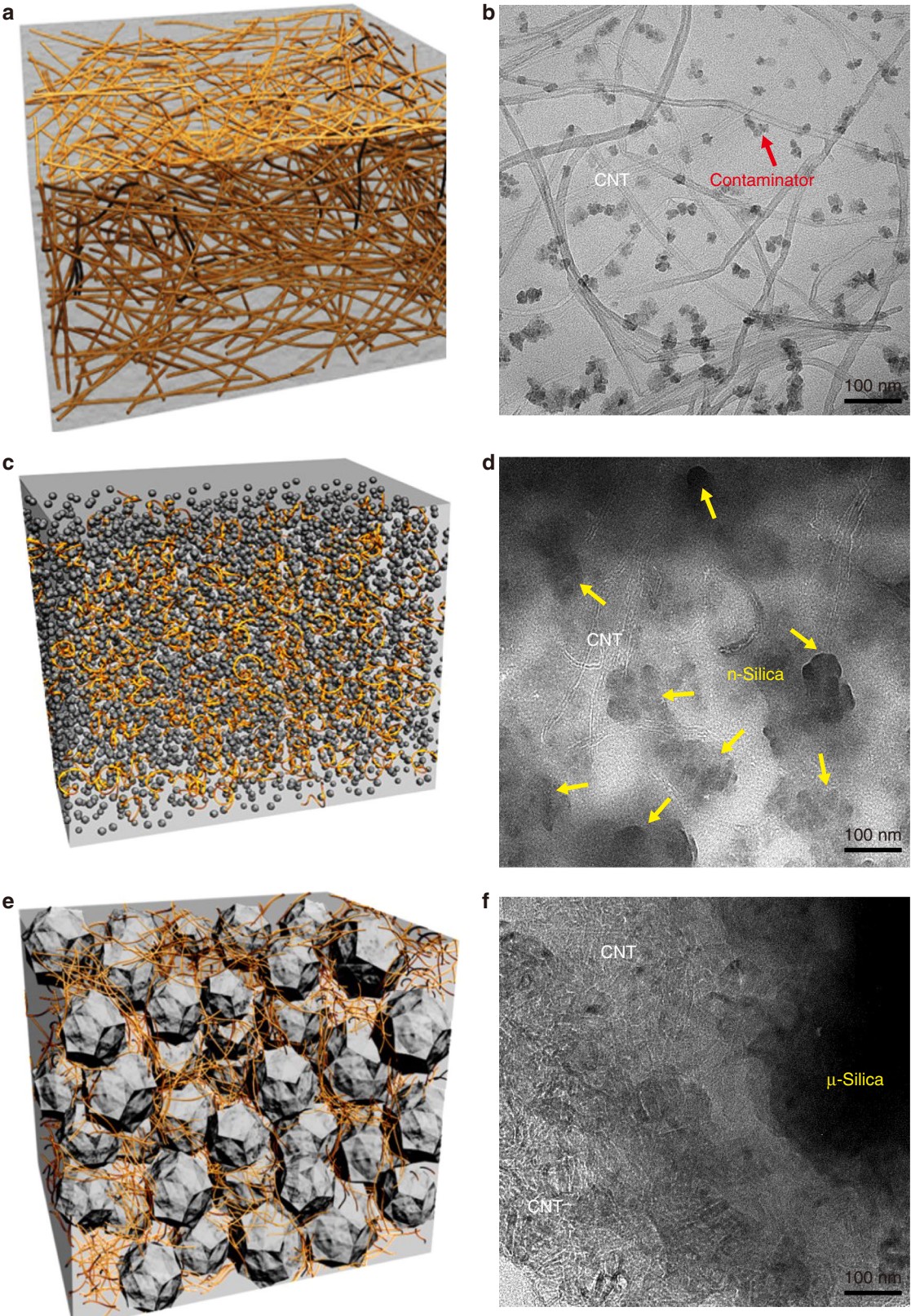

**Fig. 3** Graphical schemes and TEM images. **a**, **b** CNT composite without silica. **c**, **d** Nano-silica fillers with diameter of 20 nm. **e**, **f** Micro-silica fillers with diameter of 3–4 μm

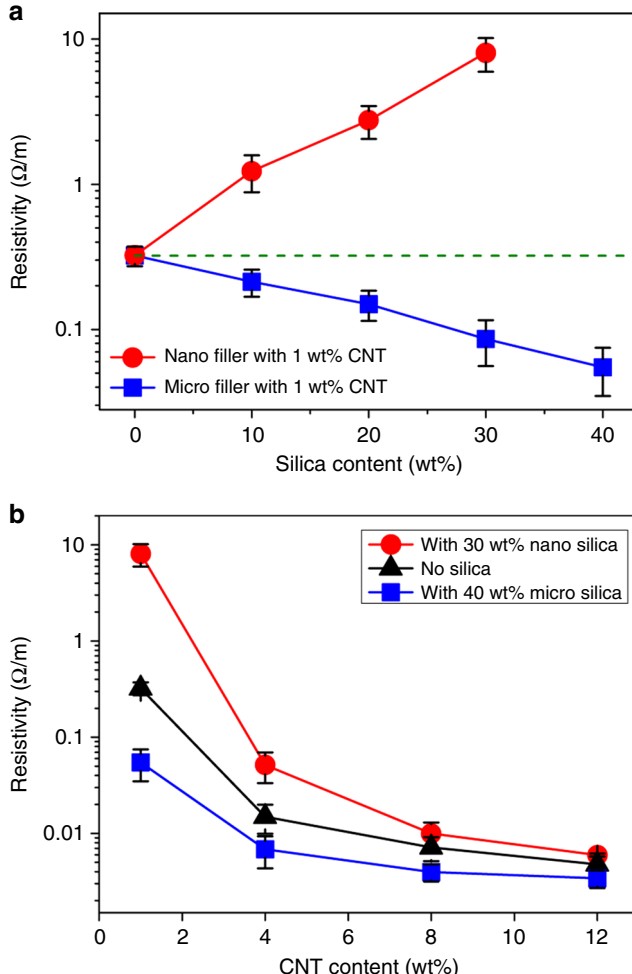

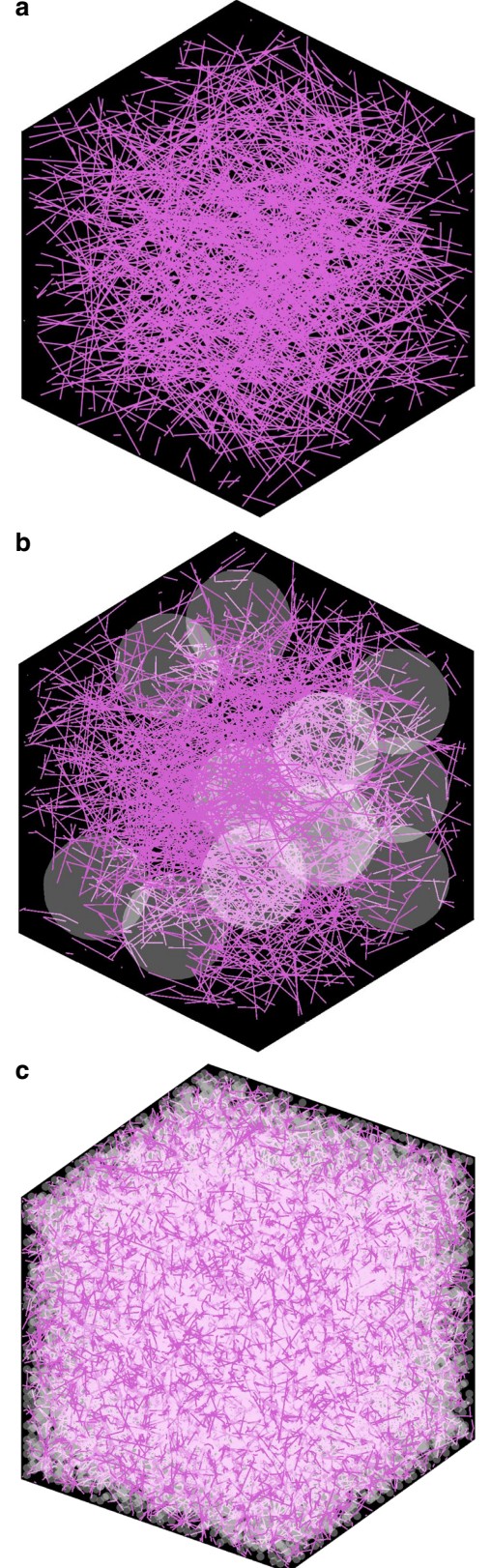

**Fig. 4** Variation of resistivity measured by experiment. **a** Resistivity of the CNT/silica composites (square: micro silica, circle: nano silica) as a function of silica wt% for fixed CNT content (1 wt%). **b** The variation of resistivity as a function of CNT wt% for fixed silica content (micro silica: 40 wt%, nano silica: 30 wt%). Error bars represent the variations of the values measured by 5 independent experiments

junctions of wires acting effectively as a conducting path in percolating clusters that span across the 3D simulation domain increases from 502 (±233) to 2599 (±438) after incorporating micro-size silica particles. The increase in the total number of interconnecting junctions in the percolating network is an explicit evidence of the resistivity reduction of the network.

On the other hand, in the CNT composite of nano-size fillers with the diameter of 50 nm, a number of curly CNTs are entangled in the network shown in Fig. 5c. Clearly, the densely-dispersed nano-size fillers cause severe bending of CNTs. The modification of the network topology leads to the decreasing probability of CNTs forming a conducting network. The calculation results show that the total number of interconnecting junctions in active percolating clusters decreases from 502 (±233) to 269 (±81). These results correspond to the resistivity increase of the CNT/nano-size silica composite measured in the experiment.

To manifest the network topology of the CNT composite system, the simulation is also carried out with 2D CNT/silica composite models of two different silica particle sizes, and the images of a random instance of the simulation are presented in Supplementary Figs. 9, 10, and 11, respectively. The results show that analogous trends of the network topology can be observed in the 2D models.

**Fig. 5** Simulation images of CNT networks. **a** No silica particle is contained. **b** Micro-size sphere (diameter of 3 μm). **a** and **b** are the same domain size 10 μm × 10 μm × 10 μm. **c** Nano-size sphere (diameter of 50 nm). The domain size presented in **c** is 2 μm × 2 μm × 2 μm

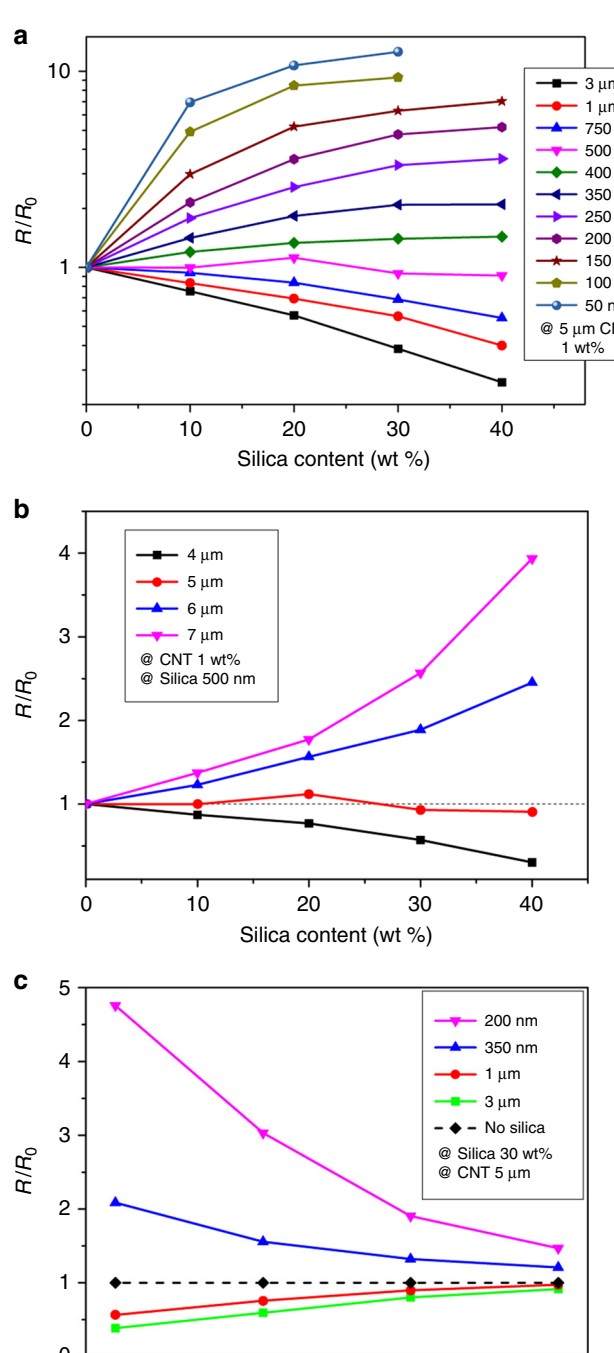

**Fig. 6** Variation of resistivity measured by simulation. **a** Simulated normalized resistance ($R/R_0$) as a function of silica wt% for different silica size. **b** $R/R_0$ as a function of silica wt% for different CNT lengths. **c** $R/R_0$ as a function of CNT wt% for different silica size with the CNT length and silica content fixed. Horizontal line indicates CNT composite without silica particle

Figure 6a illustrates the resistance change of the CNT network resulting from the modification of the network topology with 1 wt% of 5 μm CNTs. The normalized resistance is defined as $R/R_0$ to compare the average resistance of the composite systems with and without silica fillers. In the simulation, $R_0$ corresponds to $3.6 \times 10^4\ \Omega$. For a fixed amount of silica fillers, the composites of incorporating fillers with large diameters have the advantage of a

low resistance. In addition, for the case of particulate fillers from a large-diameter regime above a certain threshold value (500 nm in this case), the resistance of CNT networks decreases as the amount of incorporated silica increases. By contrast, in the case of particulate fillers from a small-diameter regime, the resistance of the CNT network increases as more silica particles are incorporated in the composite. Figure 6b plots the normalized resistance with the diameter of silica particles fixed to 500 nm and 1 wt% of the CNT content, to see how the resistance changes for increasing silica content under the CNT length variation. The resistance of the system increases for the CNT lengths above the threshold value of 5 μm CNT, while decreasing as the silica content grows for the region of the length below it. Furthermore, Fig. 6c depicts how the resistance changes as the CNT content increases for various configurations of silica diameters with the CNT length fixed. Note that $R_0$ changes with respect to the content of CNT, and their values are $3.6 \times 10^4\ \Omega$, $7.8 \times 10^2\ \Omega$, $1.8 \times 10^2\ \Omega$, and $7.1 \times 10^1\ \Omega$, for 1–4 wt%, respectively. There are consistent trends in the resistance change albeit different change rates. Thus, the resistance change of the composite system resulting from the addition of second fillers depends on the relative dimension of the CNT length and silica filler diameter.

A unified computational modeling framework for explaining the trend of the resistance observed in experiment and simulation is developed based on the combination of percolation theory[41] and Voronoi geometry[42] resulting from Swiss cheese model[43]. The system can be viewed as a novel extension of Swiss cheese model, which corresponds to a porous medium model where insulating particles occupy pores. The conductance is associated with continuum percolation[35]. However, this system is nontrivial in that the wires which the current can flow through are overlaid in the continuum-percolating region in contrast to ordinary Swiss cheese model where the current can flow freely within the continuum-percolating region. Therefore, we analyze this system as a CNT random network formed upon Swiss cheese model. For clarity, we discuss the system using a 2D model instead of a 3D model since the computational results about the 2D model show analogous percolating behaviors to the 3D model, and the model can be visualized and understood more intuitively. Nevertheless, all notions addressed here can be straightforwardly extended to the 3D model. We first consider the aspect of the continuum percolation in Swiss cheese model. The percolation in the model is characterized by Voronoi geometry[37]. A Voronoi tessellation associated with a filler is defined as the set of points closer to the center point of the filler than to centers of all other fillers in the system. The sides (corresponding to edges in 2D and faces in 3D cells, respectively) of a Voronoi tessellation constructed about the center of a spherical filler can be thought of as conducting paths around. In ordinary Swiss cheese model, the conductance of a conducting path is characterized by the width of a neck which is measured by the distance between the surfaces of two neighboring spherical fillers. It is apparent that the width of the neck is proportional to the size of the fillers. As compared to ordinary Swiss cheese model, where the total conductance depends mainly on the width of the narrowest neck, the conducting property of this system differs since the CNT network overlaid across the neck of the Voronoi tessellations can act as conducting paths. In fact, the number (or density) of CNTs that cross the neck determines the conductance of the associated side. Although wide necks exist in the system, only a small number of CNTs crossing those necks results in low conductance. We proceed to discuss all aspects of conductance properties of micro- and nano-fillers in this unified framework.

Voronoi tessellations constructed based on the center points of silica fillers (drawn in red solid lines shown in Fig. 7) correspond to possible current paths in the composite system, where

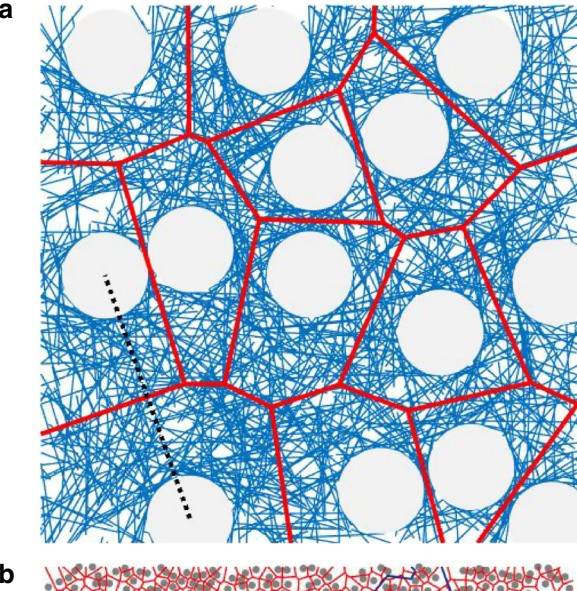

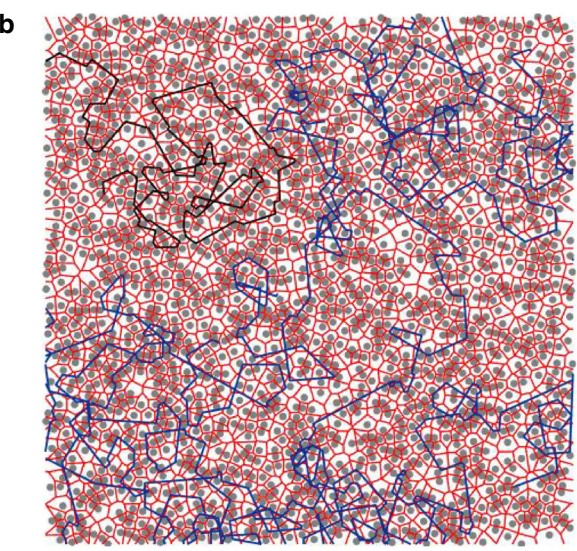

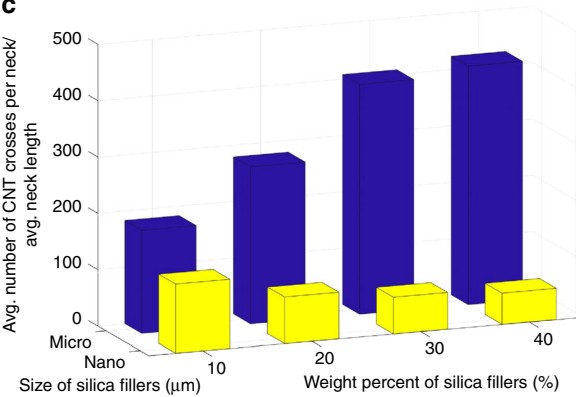

**Fig. 7** Voronoi tessellations (represented in red solid lines). Voronoi geometries are constructed based on center points of silica fillers (gray circles) **a** with a diameter of 4 μm and **b** with a diameter of 20 nm over the 2D model illustrated in Supplementary Fig. 10(b) and (d), respectively. A straight line connecting the center points of the two nearest silica fillers (represented in black dashed line in **a**) in the normal direction of a red conducting path (the side of a Voronoi tessellation) is associated with a neck in the 2D model. **c** Average number of CNT crosses per neck/average neck length for micro- and nano-silica composites with respect to silica contents

insulating particles are distributed at random in uniform transport medium. In addition, the resistivity per unit length of the current path is derived from the number of CNTs that cross a straight line (a flat surface in the 3D model) in the normal direction of the path, which connects the center points of the two nearest silica fillers (drawn in black dashed line in Fig. 7a). With this framework, the CNT network beyond a percolating limit can be described as continuum transport medium with one representative current path, and incorporating silica fillers to the CNT network leads to the addition of the current paths along with the increase of the average resistivity of the conducting paths.

The total resistance of the composite system can be determined by considering two factors: the number of the current paths and the average resistivity of the paths. To be specific, a current path is formed by connecting sides of Voronoi tessellations such that it traverses the simulation domain, and the number of such paths is proportional to the amount of the current flowing across the domain. In general, the longer lengths of the tessellation sides that form the path are, the less number of contacts among CNTs are passed, indicating that the smaller number of contact resistors exist in a current path. Furthermore, the resistivity of a side in a Voronoi tessellation is inversely proportional to the number of its crossing CNTs. Since each side can be included in a traversing current path, the average of its resistivity characterizes the average total resistance. Thus, the total resistance of the composite system decreases as the number of current paths increases, while the total resistance increases as the average resistivity of the paths increase. However, the impacts of two factors on the resistance change of the composite differ depending on the condition of the composite system. For example, when the number and resistivity of the paths are small as shown in Fig. 7a, the change in the number of the current paths mainly affects the change in the total resistance. On the other hand, when the number and resistivity of the paths are large as shown in Fig. 7b, the change in the average resistivity of the paths has a predominating effect on the change in the total resistance.

Under the fixed CNT content, the addition of micro-size silica fillers (Fig. 7a) to the CNT network results in the total resistance reduction of the CNT/silica composite due to the increment of the current paths. In this case, although the average resistivity of the current paths increases, the addition of the paths further reduces the total resistance of the composite system. The addition of the same amount of nano-size silica fillers (Fig. 7b), however, results in the increase of the total resistance of the CNT/silica composite. In this case, although a number of the representative current paths are interconnected to establish a well-conductive network, the average resistivity of the paths becomes significantly high so that the total resistance of the composite increases after incorporating the nano-size silica fillers. Note that, in the case of silica fillers with the diameter of a certain threshold value (about 500 nm in Fig. 6a), the total resistance of the composite system remains almost unchanged, implying that the effects of the current path addition and the resistivity increment of the paths are comparable.

The aspects of change in the total resistance of the composite system for increasing silica content are also explained in a consistent way with the two factors. According to the experiments and simulation results shown in Figs. 4a and 6a, respectively, the total resistance of the CNT/micro-size silica composite decreases as the content of the micro-size silica fillers increases. For this composite system, where the change in the number of current paths is a dominant factor in determining the total resistance change of the CNT/silica composite, the total resistance decreases as the content of silica particle increases due to the increase of the number of current paths in the composite system. For the CNT/nano-size silica composites, however, the total resistance

increases as the silica content increases. In this case, the change in the average resistivity of the paths is dominating, and thus the total resistance of the CNT/silica composite increases as the content of silica particle increases due to the increase in the resistivity of the current paths. For the CNT/silica composites with silica fillers of a certain value of the diameter, the total resistance of the composite system is almost fixed with increasing silica content. The effects of the current path addition and the average resistivity increment of the paths occurred by the addition of the silica particles are comparable, thereby canceling out these opposite effects.

The impacts of the number of current paths and the average resistivity of the paths on the total resistance of the composite system can be characterized quantitatively as illustrated in Fig. 7c. The number of current paths has a proportional relationship with the average number of CNTs that cross a neck formed between two neighboring fillers. To see this more carefully, the number of silica fillers is considered in the composite system. As the number of fillers increases, the number of necks subsequently increases. For a fixed number of CNTs in the composite system, only a subset of necks have CNTs crossing them, and only sides of Voronoi tessellations associated with those necks can contribute to the current paths of the percolating network. Thus, the average number of CNT crosses per neck can be used for examining the number of current paths. On the other hand, the average resistivity of a current path can be reinterpreted as the density of CNTs along the sides of Voronoi tessellations since a larger number of CNTs aligned along the side of a Voronoi tessellation can act as a more efficient conducting path. Furthermore, the density of CNTs along the side can be quantified by the density of CNTs that cross the neck associated with the side, which is evaluated by the number of CNT crosses divided by the length of the neck. To address both quantitative relationships simultaneously, the ratio of the average number of CNT crosses per neck to the average neck length is introduced as a new figure of merit for the total resistance of the composite system, and numerical results for various configurations are plotted in Fig. 7c. For four different silica content configurations of 10–40 wt%, values of the performance metric are compared for micro-silica and nano-silica cases. For clarity, the results for nano-silica fillers are boosted by 150 times since their values are too small compared to the corresponding counterparts with micro-silica fillers. A consistent trend that the metric value of micro-silica fillers exceeds that of nano-silica fillers for a given CNT content indicates that the micro-silica composite system has a higher conductance than the nano-silica composite system. In addition, the contrasting trends of the performance metric succinctly represent the opposite behaviors of the network resistance for two contrasting filler configurations. These results shed a light on the single-parameter characterization of generalized composite systems.

In summary, we have shown that the conductivity variation of CNT composites containing secondary fillers depends on the filler size, which has been confirmed through the combination of Voronoi geometry and percolation theory. The combined model has made a success in explaining the effect of the dimension of second fillers on the electrical conductivity on fiber base composite systems. These achievements lead to useful design guidelines for the use of the secondary filler in the fabrication of highly-conducting and reinforced CNT composites.

## Methods

**Synthesis of polymer composites**. For the fabrication of nano-composites, polydimethylsiloxane (PDMS: Sylgard 184 SILICONE ELASTOMER BASE) was purchased from Dow Corning. 10–15 μm long multi-walled carbon nanotubes (MWCNTs) with 15 nm outer diameter were purchased from Hanwa Nanotech Inc. (CM 250). Silica powders of 3–7 μm and 10–20 nm particle size were purchased from Silbond (Quarzwerke) and Sigma Aldrich, respectively. To ensure effective mixing and dispersion of entangled CNTs and silica particles within the polymer matrix, premixing (Paste mixer, DAEHWA TECH, PDM-lk) were used along with the three-roll milling technique (three-roll mill, Torrey Hills Technologies). A paste was made by mixing varying amounts of CNT/silica and PDMS base elastomer. The elastomer base and curing agent were combined in a 10:1 (w/w) ratio and then mixed with CNT/silica for 1 min to form the CNT/silica paste. Then, the CNT/silica pastes were three-roll milled for several minutes while gradually reducing the gap between the rolls. For the curing process, a hot press (Carver M) was used to press the CNT/silica pastes into the desired thickness at 120 °C for 30 min under a constant pressure of 2 tons. A CNT/silica/PDMS composite thickness of 1 ± 0.05 mm was used for all measurements reported in this paper.

**Morphological imaging and electrical measurement**. The morphology and structural features of CNT/silica/PDMS composites were characterized through SEM (Phillips XL30) and TEM (TECNAI F20). SEM samples were quenched in liquid nitrogen and broken to achieve fracture surfaces. Typically, the surfaces were coated with a thin Au/Pd over layer (5–10 nm in thickness) to increase the imaging contrast. In addition, thin cross-sections of the CNT polymer composites were prepared for TEM observation by ultramicrotomy after embedding them in epoxy resin. For measurement of the conductivity, the composites were treated with oxygen plasma (Oxford Plasmalab 80 RIE) prior to electrically contacting the surface. Subsequently, gold was sputtered on at the thickness of 50 nm to create contacts. The four-wire resistance method was used to measure the resistance ($R$) for composites with $R < 1$ GΩ. A Keithley 487 Picoammeter and a Keithley 2400 Sourcemeter were used for these tests. For the measurement, samples with a rectangular cross-section (5 mm wide × 0.5 mm thick, typically) were used. The outer current leads were separated by 25 mm while the inner voltage leads were separated by 15 mm. Based on the dimension, the resistivity of composite was calculated ($\rho = R \times A/l$ where $\rho$ is resistivity, $R$ is the electrical resistance, $A$ is the cross-section, and $l$ is the length of the sample). We assume that current flows through the bulk of the sample due to sufficient surface treatment and gold electrode.

**Computational modeling**. To represent a random configuration of silica particles on a cubic domain, spheres are scattered by choosing the center coordinates of each individual sphere uniformly at random. Spheres are placed one-by-one so that the distances from the designated center position to the center positions of pre-existing spheres are apart from each other farther than the diameter. The diameter of the sphere is set to the average size of the silica filler, and the number of spheres per unit volume is determined according to the weight ratio of silica particles in the composite samples used in the experiments.

Upon the completion of the random placement of circular fillers, a CNT network is created on the domain by placing multiple random instances of a flexible wire. According to morphological characteristics as observed in the SEM and TEM images, a random instance of the wire is placed so that a CNT wire is distributed uniformly in the simulation domain and can proceed forwardly to pass through a gap between silica particles but cannot penetrate the particles. The flexibility of a CNT wire is reflected by modeling them as a series of small line segments connected with joints that can bend freely within a certain degree (up to 120°). To create a single wire on the domain, constituting line segments are placed in sequence from one side to the other. The first line segment is placed at random in the volume that spheres do not occupy. Subsequent line segments are placed in a straight line unless they touch spheres. If the line segment touches a sphere, the direction of the line segment is changed toward a tangential direction of the sphere in contact so that the wire smoothly circumvents the sphere. The width and length of a single wire are set to the average diameter and length of CNTs used in the experiment, respectively. The length of a line segment is set the same as the width of the wire, and thus the line segment can be thought as of a small spherical cylinder of the identical height and width, i.e., with a square side view. This facilitates the test for the connectivity of two wires since the contact suffices to check whether, at each endpoint of a line segment, there exists any other endpoint of a line segment from a different wire within a ball of radius equal to the length of the line segment centered at the endpoint. A graphical description of the procedure is illustrated in Supplementary Fig. 1. A sample cubic domain of 10 μm × 10 μm × 10 μm is taken from the composite model to obtain the CNT network. The connectivity of the domain is checked by examining the contacts between flexible wires. For a pair of two wires, all possible pairwise distances between two line segments from respective wires are calculated. If at least one value of the distances is less than 2 nm, which is based on a tunneling mechanism (see Supplementary Note 1)[44–47], the two wires are considered to be in contact at the position of the corresponding line segments. A graphical representation of the procedure is presented in Supplementary Fig. 2. A clustering analysis identifies percolating clusters of wires that span across the domain. Starting from the leftmost wires touching the left end, each cluster is expanded gradually toward the right end by including wires in contact. If a cluster contains one of the rightmost wires touching the right end, this cluster is declared to be a conducting path. For all conducting paths, KCL is applied at all junctions of wires to construct a system of linear equations about voltage drops across the junctions due to the contact resistivity ($\rho_c$). In the simulation, $\rho_c$ is considered as a tunneling resistivity arising from the

thin insulating layer (i.e., epoxy resin) between crossing CNTs[44–47]. Since 1 V voltage is assumed to apply across the domain, the inverse of the solution of the system of linear equations corresponds to the overall resistance of the network. This process is repeated to obtain 500 independent random instances, and an ensemble average of the resistance is calculated based on the set of tested samples between 30th percentile and 70th percentile. The simulation is conducted over different configurations of silica content, silica size, CNT content, and CNT dimension. For massive simulation, the source code is implemented in MATLAB parallel computing tool box[48].

## Data availability

The data that support the findings of this study are available from the authors on request.

## Code availability

The code for performing the experiments is available from the authors on request.

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

## Acknowledgements

This research was supported by the National Research Foundation of Korea (NRF) grant funded by the Korea Government (No. 2016R1C1B1012710) and Ministry of Science, ICT and Future Planning (No. 2017R1C1B5076588). This research was also supported by Korea University Future Research Grant.

## Author contributions

S.-H.P. and D.K. designed the project with initial project design. S.-H.P. carried out experiments. G.-S.P. obtained TEM image. S.L. made analysis of image data. S.H.L. derived initial mathematical modeling of the project, wrote the simulation software, and revised the software to include computational feature. J.H. and M.Z. carried out computer simulation and revised the software to accelerate the computation speed. S.H.L. and J.H. devised Voronoi geometry-based analytical approach and Swiss cheese modeling in addition to obtain the images of simulation results. D.-J.Y. and J.-H.H. provided experimental support. S.-H.P., J.H., and S.H.L. interpreted the data. S.-H.P. wrote the initial manuscript. J.H. and S.H.L. revised the manuscript. Other authors reviewed and approved it.

## Additional information

**Competing interests:** The authors declare no competing interests.

