## [Peer Review File · Nature Communications]

Reviewers' Comments:

Reviewer #1:

Remarks to the Author:

This paper develop an interesting approach to the simulation of electrical conductivity of composites supported by some experimental evidence. The basic idea is appealing and the results interesting. However, in my view, to meet journal standard a number of improvements are needed, as listed below

MAJOR REMARKS

As far as CNT are concerned a few points need to be addressed and properly implemented concerning their length and shape:

- the exact Nanocyl type of CNT must be indicated as Nanocyl products are widely different and their shape will have a critical impact on both experimental results and appropriateness of the simulation

- effect of ball milling: the ball milling might not only break the bundles but also lead to the length reduction of CNTs so that the actual dispersed CNTs are not matching datasheet values.

- either taking the value from the datasheet or from a more detailed analysis (see below) the length dispersion should be introduced in the model if proper simulation is targeted. In the model all CNTs have the same length, as far as I understood, while the reality is rather different

- the shape of CNT is linear unless a kink is needed to avoid particles. This is not the reality for most Nanocyl products where the defects lead to 2-d or even 3-d shapes of CNT.

The experimental info needed to better model these points can be extracted, even if some issues might remain about the statistical significance of the results, by SEM and TEM analysis.

The distance between CNTs (as per Fig. S2) appears to be computed among edges and not segments. As the distance between two lines in space can be calculated by a simple geometrical formula and the approach used by authors misses the possibility that segments come closer than edges I suggest this part to be reconsidered as it might impact on percolation behaviour.

In line 109 the authors use the term 'circular disks' to describe silica particles. To me a disk has a cylindrical shape while in Fig.7 they appear to be almost spherical. If the term is wrong, please correct. If it is correct please note that in 3-d configuration an additional variable is the disk axis orientation.

Although the system is a 3-d one, it appears (although it is not clearly stated, or I missed the statement) that a 2-d simulation is carried out. If my understanding is correct a major problem arise as (i) 2-d and 3-d percolation thresholds and exponents usually differ, and (ii) additional paths which might impact on resistance values will be missed.

As per fig.6 and lines 270ff 'the resistivity per unit length is derived from the number of CNT that cross a straight line ...'. This approach disregards the fact that if the crossing is not perpendicular the contribution of the charge transport to the current is lower. As the angle is known this factor can be taken into account.

In lines 146ff the contact resistance is 'considered as a tunneling resistance'. If this is the case a value for the tunneling parameter in the exponential formula is needed. More in general the part about the calculation of the overall resistance of the network needs more detail, either in the experimental or in the supplementary information, including the resistivity of CNT or at least the ratio between this value and the resistance per unit length assumed for R_c .

Another critical information that should be provided is about the measurement. The geometry of electrodes is not clear: are they placed in such a way that the conduction can occur on the surface or in a way that the current flows through the bulk of the samples? Either a supplementary figure or a better description are needed.

MINOR REMARKS

In some parts of the paper, including Abstract, English needs improvement.

Please check the statement of lines 126-128 which is repeated in lines 376-377.

More recent works on the electrical conduction in composite simulation have been published. The authors should check the literature and update their reference list on simulation.

The description in lines 280 ff can be made clearer.

Error bars are missing (see for instance lines 215 and 224) so that the author's statements cannot

be substantiated against them.

Following lines 194-197 the percolation behaviour is observed. The authors should provide the β exponent value and comment on it.

Following Fig.2a, the size of silica microparticles is not 3-5 μm . This can be a consequence of the three rolls milling treatment and should be considered in simulation. As per Fig.2b the magnification is too low to see anything clearly.

The caption of fig.5 can be made clearer. In Fig.5b please write 'Without Silica'.

As per fig.S4 (Raman spectra) I agree with the comments. However one of the critical points of the spectra analysis is the background subtraction which can impact on the actual signal. An additional figure (S4b) with the raw spectra might help the skilled reader to appreciate this point. Moreover information can be obtained also from the 2D region (above 2400 cm^{-1}).

The samples with high CNT contents (experimental) have reached an uniform dispersion ? Usually dispersion above 5-10 wt% is rather a challenge even with 3-rolls milling. Taking into account the amount of silica, much higher wt% with respect to polymer have been reached.

ADDITIONAL COMMENTS

The fact that 250 nm size is the 'no impact value' is related to the length of CNT ? Have the authors attempted to vary the CNT length and simulate the system again ?

More than with the wt% the impact of CNT filling on conductivity (especially below the percolation threshold but not only) correlates with the number of CNT per unit volume and their length. This points can be kept in mind if and when other CNT geometries will be tested.

Reviewer #2:

Remarks to the Author:

Through the combination of simulation and experimental study, the authors investigate the size effect of the secondary particulate fillers on the electrical conductivity of CNT polymer composites. Two opposite tendencies are observed in their study, i.e., for CNT composites with micro-sized second fillers, the conductivity is enhanced, while for those with nano-size silica fillers, the conductivity is decreased. They interpret these phenomena through the combination of Voronoi geometry and percolation theory. Although I think this is a valuable contribution to the scientific community, this work is still not so qualified for being published in Nature Communications. The following points are raised to help the authors to improve their whole manuscript, which, then, could be more suitable to be submitted to other journals.

1) In their simulation model, the CNT wires are allowed to bend freely within a certain degree. What is the criteria? How does the authors connect it with the real situation?

2) In CNT polymer composites, the CNTs are often separated by insulated polymer matrix. Only when the CNTs are close enough, a pathway for carrier can be formed due to the tunneling effect. In the present study, the authors defined this distance to be one wire width but did not give any explanation. What is the reason for this choice? The authors are advised to justify this based on the experimental results.

3) In their simulation study, the authors claimed that the densely-dispersed nano-sized particulate fillers cause severe bending of CNTs, as presented in the Figure 4 (d), which results in the disruption of CNT networks. While this result is of interest, the authors did not do an experimental characterization to confirm this prediction.

4) Minor points:

- "Carbon nanotube" in the first line of the abstract should be "carbon nanotube"

RESPONSE LETTER

Reviewer #1: *This paper develops an interesting approach to the simulation of electrical conductivity of composites supported by some experimental evidence. The basic idea is appealing and the results interesting. However, in my view, to meet journal standard a number of improvements are needed, as listed below*

MAJOR REMARKS

COMMENT:1. *As far as CNT are concerned a few points need to be addressed and properly implemented concerning their length and shape: The exact Nanocyl type of CNT must be indicated as Nanocyl products are widely different and their shape will have a critical impact on both experimental results and appropriateness of the simulation*

RESPONSE: In fact, we have used two types of CNTs purchased from Nanocyl (NC 7000) and Hanwa Nanotech Inc (CM 250) for various configurations of experiments. NC 7000 is an industrial purpose CNT (the minimum order of 1 kg for purchase) leading to wide distribution of the length, while CM250 is usually of research purpose with relatively concentrated length distribution. To obtain the consistency of results of experiments, 10-15 μ m long Hanwa CNTs (CM 250) of ~15 nm outer diameter were selected. This

information has been added to the revised manuscript. The detailed discussion about their shape/dimension change due to three-roll milling is addressed in the response of the following comment.

COMMENT:2. *Effect of ball milling: the ball milling might not only break the bundles but also lead to the length reduction of CNTs so that the actual dispersed CNTs are not matching datasheet values.*

RESPONSE: From the literature, the reduction of the length of CNTs depends on three-roll shear intensity and time [R1][R2]. For example, the length distribution of as-received CNTs is 10-15 μm . To examine actual length of the CNTs after three roll milling, methyl isobutyl ketone (MIBK) was used to dissolve uncured polymer resin in the CNT paste. Bath type sonication (Branson) was used to disperse the CNTs and the diluted solution was dropped on a silicon wafer. SEM was used to measure the length. The range of the CNT length after three-roll milling becomes 3-6 μm (based on 80-100 CNT sample measurements) as shown in Figure R1, while 4 μm long CNTs were measured most frequently. Images in Figure R2 are typical images of CNT before/after three-roll milling. Based on these facts, the length of CNT was chosen as 4 μm for the computational efficiency of simulation. This has been added into the revised manuscript and Supplementary Information.

Figure R1. Distribution of CNT length before and after three-roll milling

[R1] I.D. Rosca, S. V. Hoa. Highly Conductive Multiwall Carbon Nanotube and Epoxy Composites Produced by Three-Roll Milling. Carbon 47 (2009) 1958-1968.

[R2] S. Fu, Z. Chen, S. Hong, C. C. Han. The reduction of carbon nanotube (CNT) length during the manufacture of CNT/polymer composites and a method to simultaneously determine the resulting CNT and interfacial strengths. Carbon 47 (2009) 3192-3200

Figure R2. SEM of CNT before and after three-roll milling

COMMENT:3. *Either taking the value from the datasheet or from a more detailed analysis (see below) the length dispersion should be introduced in the model if proper simulation is targeted. In the model all CNTs have the same length, as far as I understood, while the reality is rather different.*

RESPONSE: As shown in **Figure R1** and **R2**, the CNT length information after three-roll milling were specified in the manuscript and Supplementary Information. In addition, after three-roll milling, the length of a CNT is distributed over 4-5 μm (mostly 4 μm). In this respect, we assume that the average length of a CNT is 4 μm in simulation.

COMMENT:4. *The shape of CNT is linear unless a kink is needed to avoid particles. This is not the reality for most Nanocyl products where the defects lead to 2-d or even 3-d shapes of CNT.*

RESPONSE: We have observed in extensive experiments that CNTs could exist linearly in the composite system. In fact, we have used two types of CNTs purchased from Nanocyl (NC 7000) and Hanwa Nanotech Inc (CM 250) for various configurations of experiments. NC 7000 is an industrial purpose CNT (the minimum order of 1 kg for purchase) leading to wide distribution of the length, while CM250 is usually of research purpose with a relatively concentrated length distribution. To obtain consistent experimental results, Hanwa CNTs (CM 250) were used. We have corrected the corresponding information in the revised manuscript. With Hanwa CNTs in the CNT/PDMS composite with no silica particles, the shape of CNTs exist in a relatively linear shape. To ensure this claim, we took transmission electron microscopy (TEM: TECNAI F20) images of the CNT/PDMS composites. Thin cross-sections of the CNT polymer composites were prepared for TEM observation by ultramicrotomy after embedding them in an epoxy resin. Figure R3 (b) shows that CNTs often exist in a linear shape. The corresponding comments have been added to the revised manuscript.

(a)

(b)

Figure R3. (a) Scheme and (b) TEM image of CNT/PDMS composite without silica fillers

COMMENT:5. The experimental info needed to better model these points can be extracted, even if some issues might remain about the statistical significance of the results, by SEM and TEM analysis.

RESPONSE: We have carried out intensive SEM and TEM analysis. First, we have measured SEM images for all cases, which show clear differences in morphological distribution as shown in Figure R4. This has been added to the revised manuscript.

In addition, to help understanding of TEM images, we have presented scheme images together in Figure R5. In case of no silica ((a) and (b)), CNTs exist in a relative linear shape. By contrast, in case of nano-silica composites (d), twisted CNTs were observed. In addition, in micro-silica composites (f), CNTs were concentrated to avoid micro-silica particles. Note that, during ultramicrotomy sectioning, the surface roughness of cross sections caused by silica particles results in blurs in images (d) and (f), which ends up with relatively less clear images than (b) corresponding to CNT/PDMS only case. This has been added to the revised manuscript.

Figure R4. SEM images of CNT composite (a) without silica, (b) nano silica (~20 nm) and (c) micro silica (3~4 μm)

Figure R5. Graphical scheme and TEM image of CNT composite without silica case ((a) and (b)), nano silica (~20 nm) case ((c) and (d)), and micro silica (3~4 μm) case ((e) and (f))

COMMENT:6. The distance between CNTs (as per Fig. S2) appears to be computed among edges and not segments. As the distance between two lines in space can be calculated by a simple geometrical formula and the approach used by authors misses the possibility that

segments come closer than edges I suggest this part to be reconsidered as it might impact on percolation behaviour.

RESPONSE:

To avoid the confusion about the geometry pertaining to percolation network elements, we have revised the figure that explains the overlap check of CNTs (Figure R6, corresponding to Figure S2 in Supplementary Information). Since the length of a segment is chosen to equal to the width of CNT, one segment corresponds to a square with the length of the side equal to the width of a CNT. This ensures that the distance of the closest points of two segments from distinct CNTs is equivalent to the distance of the closest end points of the segments. Therefore, all mathematical techniques that we have considered to check the contact of multiple CNTs turn out to be valid. We believe that possible cases of the contact are properly considered in the current model.

Figure R6. How to determine whether two nanowires have a contact

COMMENT:7. In line 109 the authors use the term ‘circular disks’ to describe silica particles. To me a disk has a cylindrical shape while in Fig.7 they appear to be almost spherical. If the term is wrong, please correct. If it is correct, please note that in 3-d configuration an additional variable is the disk axis orientation.

RESPONSE:

The Monte-Carlo simulation has also been performed in the 3-D configuration along with the 2D configuration as presented in the simulation images shown in Figure R7. In fact, a circle that represents a silica particle is not a cylinder in the 3D domain. Instead, the circle corresponds to a spherical object of radius equal to the average size of a silica particle. In the 3-D simulation, spheres and bendable wires respectively, are randomly placed in a cubic domain. We see that the results obtained from the 3-D simulation are consistent with the experimental data as presented in Figure R8

Figure R7. Simulation images of CNT networks (a) without silica, (b) with micro-size sphere (filler diameter = $3\ \mu\text{m}$), and (c) with nano-size sphere (filler diameter = $50\ \text{nm}$). (a) - (b) are same domain size ($10\ \mu\text{m} \times 10\ \mu\text{m} \times 10\ \mu\text{m}$), and domain size of (c) is $2\ \mu\text{m} \times 2\ \mu\text{m} \times 2\ \mu\text{m}$.

Figure R8. (a) Simulated normalized resistance change (R/R_0) as a function of silica wt% for different silica size. The values are matched to experimental results with CNT contents fixed at 1 wt% (b) R/R_0 as a function of silica wt% for different CNT lengths. (c) R/R_0 as a function of CNT wt% for different silica size with fixed silica content (30 wt%). Horizontal line indicates CNT composite without silica particle.

COMMENT:8. Although the system is a 3-d one, it appears (although it is not clearly stated, or I missed the statement) that a 2-d simulation is carried out. If my understanding is correct a major problem arise as (i) 2-d and 3-d percolation thresholds and exponents usually differ, and (ii) additional paths which might impact on resistance values will be missed.

RESPONSE:

To clarify the results, we have also performed the simulation using a 3D percolation model. According to the revised computational results, exponents of 2D and 3D models may differ. In addition, the percolation thresholds are out of scope since we desire to measure the resistance which can be found only for the after-threshold regime of the percolation behaviors, where sufficient percolation occurs. However, in both cases, we have observed similar trends in resistance variation as seen in the updated figures.

COMMENT:9. As per fig.6 and lines 270ff ‘the resistivity per unit length is derived from the number of CNT that cross a straight line ...’. This approach disregards the fact that if the crossing is not perpendicular the contribution of the charge transport to the current is lower. As the angle is known this factor can be taken into account.

RESPONSE:

In the case of the proportion of silica fillers large enough to affect the total conductance of CNT network (about over 10%), the width of the neck between silica fillers is very small compared to the length of the CNT. In this setup, the longitudinal direction of a CNT contained in the neck does not deviate significantly from the direction of the conducting path (the edge of the Voronoi tessellation). Thus, the total current flowing through the neck between two spherical fillers can be predicted by counting the number of CNTs crossing the neck. This can also be interpreted based on the definition of electrical current, which is the amount of charges passing the cross section during unit time interval. Since the neck is associated with the cross section and charges flow through the wire, the number of wires that cross the neck is essentially proportional to the total amount of charges flowing through the cross section. Thus, it is a valid strategy to characterize the electrical current by the number of wires crossing the neck.

COMMENT:10. In lines 146ff the contact resistance is ‘considered as a tunneling resistance’. If this is the case a value for the tunneling parameter in the exponential formula is needed. More in general the part about the calculation of the overall resistance of the network needs more detail, either in the experimental or in the supplementary information, including the resistivity of CNT or at least the ratio between this value and the resistance per unit length assumed for R_c .

RESPONSE:

For 3-D simulation, the tunneling resistivity (ρ_c) between CNTs is estimated by [R3]

$$\rho_c = \frac{h^2}{e^2 \sqrt{2m\lambda}} \exp\left(\frac{4\pi d}{h} \sqrt{2m\lambda}\right),$$

where h is the Planck’s constant, e is the electric charge of electron, m is electron mass, d is the tunneling distance between CNTs, and λ is the tunneling barrier height which set to 0.5 eV. The maximum value of the tunneling distance d is assumed to be 6 nm implying that two CNTs closer than 6 nm are assumed to be electrically connected. [R4-R7] The contribution of CNT resistivity is considered to be negligible as compared to the contact resistance, thereby not considered in the simulation. [R8] In addition, we have added more detail about the calculation of the overall resistance in the revised manuscript.

[R3] Hu, N., Karube, Y., Yan, C., Masuda, Z., & Fukunaga, H. Tunneling effect in a polymer/carbon nanotube nanocomposite strain sensor. *Acta Mater.* 56, 2929-2936 (2008).

[R4] Li, J., Ma, P. C., Chow, W. S., To, C. K., Tang, B. Z., & Kim, J. K. Correlations between percolation threshold, dispersion state, and aspect ratio of carbon nanotubes. *Adv. Funct. Mater.* 17, 3207–3215 (2007)

[R5] Ruschau, G. R., Yoshikawa, S., & Newnham, R. E. Resistivities of conductive composites. *J. Appl. Phys.* 72, 953-959 (1998)

[R6] Zare Y. & Rhee, K. Y. A simple methodology to predict the tunneling conductivity of polymer/CNT nanocomposites by the roles of tunneling distance, interphase and CNT waviness. *RSC Adv.* 7, 34912-34921 (2017)

[R7] Zou, J. -F., Yu, Z. -Z., Pan, Y.-X., Fang, X.-P., & Ou, Y.-C. Conductive mechanism of polymer/graphite conducting composites with low percolation threshold. *J. Poly. Sci.: Part B: Poly. Phys.* 40, 954–963 (2002)

[R8] Yu, Y., Song, G. & Sun, L. Determinant role of tunneling resistance in electrical conductivity of polymer composites reinforced by well dispersed carbon nanotubes, J. Appl. Phys. 108, 084319 (2010)

COMMENT:11. Another critical information that should be provided is about the measurement. The geometry of electrodes is not clear: are they placed in such a way that the conduction can occur on the surface or in a way that the current flows through the bulk of the samples? Either a supplementary figure or a better description are needed.

RESPONSE:

The composites were treated with oxygen plasma (Oxford Plasmalab 80 RIE) prior to electrical contacting of the surface. Subsequently, 50 nm of gold were sputtered on for the contacts. The four-wire resistance method was used to measure the resistance (R) for composites with $R < 1 \text{ G}\Omega$, using the Keithley 487 picoammeter and the Keithley 2400 Sourcemeter. For measurement, we used rectangular cross-sections of samples (5 mm wide \times 0.5 mm thick, typically). The outer current leads were separated by 25 mm, while the inner voltage leads were separated by 15 mm. Base on such configurations of the dimension, the resistivity of composite were calculated ($\rho = R \cdot A / l$ where ρ is resistivity, R is the electrical resistance, A is the cross-section are of sample and l is the length of the sample). To do so, we assume that electrical current flows across the cross-section of the sample due to sufficient surface treatment and gold electrode. This has been added to the revised manuscript.

MINOR REMARKS

COMMENT:12. In some parts of the paper, including Abstract, English needs improvement.

RESPONSE:

We have made efforts to improve the English use as suggested.

COMMENT:13. Please check the statement of lines 126-128 which is repeated in lines 376-377.

RESPONSE:

We have rephrased the corresponding statements and revised to provide the information about details of the simulation.

COMMENT:14. More recent works on the electrical conduction in composite simulation have been published. The authors should check the literature and update their reference list on simulation.

RESPONSE:

We have updated the following additional references in the Introduction:

[R9] Ni, X., Hui, C., Su, N., Jiang, W., & Liu, F. Monte Carlo simulations of electrical percolation in multicomponent thin films with nanofillers. *Nanotechnology* **29**, 075401 (2018)

[R10] Jin, L., Chortos, A., Lian, F., Pop, E., Linder, C., Bao, Z., & Cai, W. Microstructural origin of resistance-strain hysteresis in carbon nanotube thin film conductors. *Proc. Natl. Acad. Sci.* 201717217 (2018)

[R11] Huang, Y., Wang, W., Zeng, X., Guo, X., Zhang, Y., Liu, P., Ma, Y., & Zhang, Y. Effects of the filler size on the electrical percolation threshold of carbon black-carbon nanotube-polymer composites. *J. Appl. Polym. Sci.* **135**, 46517 (2018)

COMMENT:15. The description in lines 280 ff can be made clearer.

RESPONSE: We have rephrased the corresponding description to provide the information about details of the simulation. A current path is formed by connecting the sides of Voronoi tessellations to traverse the simulation domain. Furthermore, the resistivity of a side of a Voronoi tessellation is inversely proportional to the number of its crossing CNTs. Since each side can be included in a traversing path, the average of its resistivity characterizes the average resistivity of the paths. Thus, their overall behavior can determine the total resistance of the composite system. We have updated the corresponding part in the revised manuscript.

COMMENT:16. Error bars are missing (see for instance lines 215 and 224) so that the author's statements cannot be substantiated against them.

RESPONSE:

It is not very straightforward to represent the corresponding information in a graphics with error bars. Instead of plotting them for the average number of interconnecting junction points, we have added its standard deviation in the revised manuscript.

COMMENT:17. Following lines 194-197 the percolation behaviour is observed. The authors should provide the β exponent value and comment on it.

RESPONSE: From the best linear fit of the data σ_{DC} to log-log plots of the power law, β was estimated in CNT composite with nano-silica as $\beta \sim 2.8$, in CNT composite without silica as $\beta \sim 1.9$, and in CNT composite with micro-silica as $\beta \sim 1.5$, respectively. We have checked from previous literature [R12-R20] that β value mostly has a range between 1.3 and 4.7, which the estimated β values fall into for a thick-film resistor configuration with elongated fibers. This has been added to the revised manuscript.

[R12] F. Du RCS, W. Zhou, S. Brand, J. E. Fischer, K. I. Winey. *Macromolecules*. 2004;37:9048.

[R13] V. Skákalová UD-W, S. Roth. *Synthetic Metals*. 2005;152:349.

[R14] E. Kymakis GAJA. *J Appl Phys*. 2006;99:084302.

[R15] Z. Liu, G. Bai, Y. Huang, F. Li, Y. Ma, T. Guo, et al. *J Phys Chem*. 2007;111:13696.

[R16] N. Li, Y. Huang FD, X. He, X. Lin, H. Gao, Y. Ma, et al. *Nano Lett*. 2006;6:1141.

[R17] B. Kim JL, I. Yu. J Appl Phys. 2003;94:6724.

[R18] P. Potschke SMD. Polymer. 2003;44:5023.

[R19] G. J. Hu CGZ, S. M. Zhang, M. S. Yang, Z. G. Wang. Polymer. 2006;47:480.

[R20] B. E. Kilbride, J. N. Coleman, J. Fraysse, P. Fournet, M. Cadek, A. Drury, et al. J Appl Phys. 2002;92:4024.

COMMENT:18. Following Fig.2a, the size of silica microparticles is not 3-5 μm . This can be a consequence of the three rolls milling treatment and should be considered in simulation. As per Fig.2b the magnification is too low to see anything clearly.

RESPONSE: We agree with the reviewer's comments. We have updated SEM images in Figure 2 with a new sample for clearer presentation. As three-roll milling process causes the decrease in CNT length, it also results in the decrease in size of micro-silica particles. For example, the size of as-received micro-silica particles is distributed over 7-10 μm . To examine actual size of micro-silica particles after three-roll milling, methyl isobutyl ketone (MIBK) was used to dissolve an uncured polymer resin (PDMS) in the CNT/micro-silica mixed paste. Bath type sonication (Branson) was used to disperse it, and the diluted solution was dropped on a silicon wafer. SEM was used to measure the size of micro-silica particles. The micro-silica size after three roll milling was found to range between 2-4 μm (based on 80-100 micro-silica sample measurement) as shown in Figure R9, where 3 μm long CNTs were measured mostly. Figure R10 depicts typical images of micro-silica before/after three-roll milling. Base on the histogram, the size of micro-silica was specified as 3 μm for simulation work. The corresponding comments have been added to the revised manuscript and Supplementary Information.

Figure R9. Distribution of micro-silica size before and after three-roll milling

Figure R10. SEM of micro-silica before and after three-roll milling

COMMENT:19. The caption of fig.5 can be made clearer. In Fig.5b please write ‘Without Silica’.

RESPONSE:

We have corrected it as indicated.

COMMENT:20. As per Figure S4 (Raman spectra) I agree with the comments. However, one of the critical points of the spectra analysis is the background subtraction which can impact on the actual signal. Additional figure (Figure S4b) with the raw spectra might help the skilled reader to appreciate this point. Moreover information can be obtained also from the 2D region (above 2400 cm^{-1}).

RESPONSE: As reviewer comments, we have added the additional figure (Figure R11) with the raw spectra as below (in the Supplementary Information)

(a)

(b)

Figure R11. Raman results (a) and (b)

COMMENT:21. The samples with high CNT contents (experimental) have reached an uniform dispersion? Usually dispersion above 5-10 wt% is rather a challenge even with 3-rolls milling. Taking into account the amount of silica, much higher wt% with respect to polymer have been reached.

RESPONSE: We have used a proficient dispersion technique using three-roll milling, in particular, in high loading CNT contents. In our previous work, we have shown well

dispersed CNTs in polymer matrix (10 wt% of CNTs) as shown below Figure R12. In addition, we have optimized the three-roll milling parameter by control the number of three roll passes as indicated in Figure R13 from [R21].

[R21] P. Theilmann, D.-J. Yun, P. Asbeck, S.H. Park, Superior electromagnetic interference shielding and dielectric properties of carbon nanotube composites through the use of high aspect ratio CNTs and three-roll milling, *Organic Electronics* 14 (2013) 1531–1537

Figure R12. SEM characterization of (a) Highly entangled raw CNT bundles, (b) Fracture image of uniformly dispersed CNT composite with high resolution (10 wt%) and (c) with low resolution (10 wt%). (vol% was converted to wt%)

Figure R13. Dependence of DC conductivity on the number of three roll passes (5.7 vol%)

ADDITIONAL COMMENTS

COMMENT:22. The fact that 250 nm size is the ‘no impact value’ is related to the length of CNT? Have the authors attempted to vary the CNT length and simulate the system again? More than with the wt% the impact of CNT filling on conductivity (especially below the percolation threshold but not only) correlates with the number of CNT per unit volume and their length. This points can be kept in mind if and when other CNT geometries will be tested.

RESPONSE:

We have performed the simulation using 3D model and the corresponding size of the CNT turns out to be 350nm, which has been updated in the revised manuscript. According to Figure R14(a), in the composite systems with 1wt% of 4µm CNTs and silica particles of diameter 350nm, the resistance remains unchanged as the silica content increases. However, Figure R14(b), which illustrates the cases with the diameter of silica particles fixed to 500nm, shows the invariance of the resistance for the 1wt% of 5µm CNT contained in the composite systems. In addition, for a larger length of the CNT around 6-7µm, the resistance of the system increases whereas, for a shorter length of 4µm, it decreases as the silica content grows. Figure R14(c) depicts how the resistance changes as the CNT content increases with the fixed CNT length for various configurations of silica diameters. It shows consistent trends in the

resistance change albeit different change rates. Thus, the change in the resistance of the composite system depends on the relative size of the CNT length and silica diameter.

Figure R14. (a) Simulated normalized resistance change (R/R_0) as a function of silica wt% for different silica size. The values are matched to experimental results with CNT contents fixed at 1 wt% (b) R/R_0 as a function of silica wt% for different CNT lengths. (c) R/R_0 as a function of CNT wt% for different silica size with the CNT length and silica content (30 wt%) fixed. Horizontal line indicates CNT composite without silica particle.

Reviewer #2: Through the combination of simulation and experimental study, the authors investigate the size effect of the secondary particulate fillers on the electrical conductivity of CNT polymer composites. Two opposite tendencies are observed in their study, i.e., for CNT composites with micro-sized second fillers, the conductivity is enhanced, while for those with nano-size silica fillers, the conductivity is decreased. They interpret these phenomena through the combination of Voronoi geometry and percolation theory. Although I think this is a valuable contribution to the scientific community, this work is still not so qualified for being published in Nature Communications. The following points are raised to help the authors to improve their whole manuscript, which, then, could be more suitable to be submitted to other journals.

COMMENT:1. In their simulation model, the CNT wires are allowed to bend freely within a certain degree. What is the criteria? How does the authors connect it with the real situation?

RESPONSE:

The rotation range of the joint between two consecutive line segments is limited to 120° based on the physical property of a CNT, in that the carbon tube structure can be preserved with very large bending angle (up to 120°) without atomic destruction due to a perfect hexagonal structure of a CNT according to [R22-R25].

[R22] Iijima, S., Brabec, C., Maiti, A., & Bernholc, J. Structural flexibility of carbon nanotubes. *J. Chem. Phys* **104**, 2089-2092 (1996).

[R23] Han, J., Anantram, M.P., Jaffe, R. L., Kong, J., & Dai, H. Observation and modeling of single-wall carbon nanotube bend junctions. *Phys Rev B* **57**, 14983-14989 (1988).

[R24] Volkov, A. N., Shiga, T., Nicholson, D., Shiomi, J., & Zhigilei, L. V. Effect of bending buckling of carbon nanotubes on thermal conductivity of carbon nanotube materials. *J Appl Phys* **111**, 053501 (2012).

[R25] Yakobson, B. I., & Avouris, P. Mechanical properties of carbon nanotubes. *Appl. Phys. Lett* **80**, 287–327 (2001).

COMMENT:2. In CNT polymer composites, the CNTs are often separated by insulated polymer matrix. Only when the CNTs are close enough, a pathway for carrier can be formed due to the tunneling effect. In the present study, the authors defined this distance to be one wire width but did not give any explanation. What is the reason for this choice? The authors are advised to justify this based on the experimental results.

RESPONSE:

In the 3-D Monte-Carlo simulation, the tunneling resistivity (ρ_c) between CNTs is estimated by [R26]

$$\rho_c = \frac{h^2}{e^2 \sqrt{2m\lambda}} \exp\left(\frac{4\pi d}{h} \sqrt{2m\lambda}\right)$$

where h is the Planck's constant, e is the electric charge of electron, m is electron mass, d is the tunneling distance between CNTs, and λ is the tunneling barrier height which set to 0.5 eV. The maximum value of the tunneling distance d is assumed to be 6 nm implying that two CNTs closer than 6 nm are assumed to be electrically connected. [R27-R30] The contribution of CNT resistivity is considered to be negligible as compared to the contact resistance that it is not considered in this simulation. [R31]

[R26] Hu, N., Karube, Y., Yan, C., Masuda, Z., & Fukunaga, H. Tunneling effect in a polymer/carbon nanotube nanocomposite strain sensor. *Acta Mater.* 56, 2929-2936 (2008).

[R27] Li, J., Ma, P. C., Chow, W. S., To, C. K., Tang, B. Z., & Kim, J. K. Correlations between percolation threshold, dispersion state, and aspect ratio of carbon nanotubes. *Adv. Funct. Mater.* 17, 3207–3215 (2007)

[R28] Ruschau, G. R., Yoshikawa, S., & Newnham, R. E. Resistivities of conductive composites. *J. Appl. Phys.* 72, 953-959 (1998)

[R29] Zare, Y. & Rhee, K. Y. A simple methodology to predict the tunneling conductivity of polymer/CNT nanocomposites by the roles of tunneling distance, interphase and CNT waviness. *RSC Adv.* 7, 34912-34921 (2017)

[R30] Zou, J.-F., Yu, Z.-Z., Pan, Y.-X., Fang, X.-P., & Ou, Y.-C. Conductive mechanism of polymer/graphite conducting composites with low percolation threshold. *J. Poly. Sci.: Part B: Poly. Phys.* 40, 954–963 (2002)

[R31] Yu, Y., Song, G. & Sun, L. Determinant role of tunneling resistance in electrical conductivity of polymer composites reinforced by well dispersed carbon nanotubes, J. Appl. Phys. 108, 084319 (2010)

COMMENT:3. In their simulation study, the authors claimed that the densely-dispersed nano-sized particulate fillers cause severe bending of CNTs, as presented in the Figure 4 (d), which results in the disruption of CNT networks. While this result is of interest, the authors did not do an experimental characterization to confirm this prediction.

RESPONSE: To demonstrate our claim, we have performed intensive SEM and TEM analysis. From the experimental images shown in Figure R15, we can see clear difference among all cases. This has been added to the revised manuscript.

In addition, to help understanding of TEM images, we have presented scheme images together in Figure R16. In case of no silica ((a) and (b)), CNTs exist in a relative linear shape. By contrast, in case of nano-silica composites (d), twisted CNTs were observed. In addition, in micro-silica composites (f), CNTs were concentrated to avoid micro-silica particles. Note that, during ultramicrotomy sectioning, the surface roughness of cross sections caused by silica particles results in blurs in images (d) and (f), which ends up with relatively less clear images than (b) corresponding to CNT/PDMS only case. This has been added to the revised manuscript.

Figure R15. SEM images of CNT composite (a) without silica, (b) nano silica (~20 nm) and (c) micro silica (3-4 μm)

Figure R16. Scheme and TEM image of CNT/PDMS composite (a), (b) without silica, (b), (d) nano silica (~20 nm) and (c) micro silica (3~4 μm)

COMMENT:4. “Carbon nanotube” in the first line of the abstract should be “carbon nanotube”.

RESPONSE:

As indicated, we have corrected it.

In conclusion, we would like to thank the reviewer again for the comments, which have all been very carefully considered and have helped improve the paper.

Best Regards,

Corresponding author

Reviewers' comments:

Reviewer #2 (Remarks to the Author):

Authors replied to review's comment properly, and the manuscript is also revised and possesses more clear discussion.

Therefore I can recommend to publish this manuscript to Nature Communications.

Reviewer #3 (Remarks to the Author):

The paper deals with a numerical investigation supported by experimental results concerning the size effect of the secondary particulate fillers on the electrical conductivity of CNT polymer composites.

The paper has been brought to my attention after a first review run. The authors have provided suitable responses to the previously issued reviewers' remarks. I agree with the other reviewers that the paper is a valuable contribution to the scientific community, but also after the first review, the paper needs further improvements to be acceptable for publication on a high-qualified journal as Nature Communications.

With respect to the present version of the paper the following further issues should be discussed by the authors.

1. The paper would benefit of a more detailed discussion on which are the novelties with respect to those available in the literature concerning the simulation of electrical conductivity of composites loaded with two types of fillers.

2. In the abstract (line 1 and ff.) the authors state "Hybrid carbon nanotube (CNT) composites comprised of more than two different types of fillers...". The statement is quite misleading since the paper deals with composites with only two types of fillers.

3. As it concerns the percolation curve (line 201 and ff.) the authors state that the composites follow a power-law characteristic and provide the values of the critical exponent β . However the reliability of the calculations is not proved, for example by providing the values of the correlation coefficient R^2 for the considered fits.

4. With regard to the CNT network the authors state (line 238 and ff.) "The calculation results show that the average number of interconnecting junctions on a single CNT decreases from 23.61 ± 0.74 to 22.58 ± 0.73 . These results correspond to the resistivity increase of the CNT/nano-size silica composite measured in experiment." The two indicated figures of interconnecting junctions, indeed, are not that different (the lower and upper values of the two ranges are overlapping) to justify the increase of the resistivity values.

5. The authors introduce (line 247 and ff) the "normalized resistance change ... defined as R/R_0 " in order to compare the average resistances of the composite systems with and without silica fillers. It is important to provide the reference value R_0 obtained for the resistance.

6. Since the main contribution of the paper is on simulation, it should provide more details on the relevant aspects concerning such aspect. In particular, information on geometrical dimensions of the considered domain have to be indicated. In fact (line 220 and ff.), the author state that "Figure 5 (a) shows a random network of CNTs with the diameter of 15 nm and the length of 4 μm leading to the resistance of 193 Ω , which is comparable to the resistance value with the experimental results shown in Figure 4". However, it is not clear if the dimensions of the simulated domain are the equal to those adopted to carry out the measurements in experimental tests.

7. As it concerns the conduction mechanism, the authors state (line 413 and ff.): "...If at least one value of the distances is less than 6nm, which is based on a tunneling mechanism (see section S8 in the Supplementary Information), the two wires are considered to be in contact at the position of the corresponding line segments."

In Sect. 8 "The maximum value of the tunneling distance d is assumed to be 6 nm implying that two CNTs closer than 6 nm are assumed to be electrically connected." The direct experience of this reviewer which is in accordance with other studies published in literature (see for example P1-P2 in the sequel) is that for a tunneling distance greater than 2nm and height of potential barrier

(λ) set at 0.5 eV, the junction is equivalent to an open circuit (the tunneling resistance is of the order of 10^{12} Ohm). In light of this strong discrepancy with the presented results, the authors should support their indications by a more detailed comparative analysis.

8. With respect to the interpretation of the resistivity of the composite (line 303 and ff.) the authors state that "The total resistance of the composite system can be determined by considering two factors: the numbers of the current paths and the average resistivity of the paths... Thus, the total resistance of the composite system decreases as the number of current paths increases, while the total resistance increases as the average resistivity of the paths increase". Such conclusions however are only qualitatively supported with Fig. 7a and b. From such figures it is quite difficult to estimate what the authors assert "... when the number and resistivity of the paths are small as shown in Figure 7 (a), the change in the number of the current paths mainly affects the change in the total resistance..." . Numerical data would help the reader to understand the authors' conclusions.

9. As it concerns the Raman Spectra introduced in Supplementary Information as Fig. S3 the authors state: "Raman spectra illustrated in Figure S3 also confirm the effect of structural changes and the defect of CNTs on the resistivity of the composites. There were no particular changes in the D-band and G-band peaks for all samples, indicating that only reversible deformation of CNTs occurred in the CNT/Silica/PDMS composites without atomic destruction." However, as discussed in the paper P3. cited in the sequel, order and defect level of carbon-based fillers can be identified by the ratio I_D/I_G rather on the observations of the two separate spectra. For CNT 1%/40% micro silica some changes are appreciable.

10. The paper still needs improvements as it concerns English; moreover there are many typos.

P1. Chunyu Li, Erik T. Thostenson, and Tsu-Wei Chou, "Dominant role of tunneling resistance in the electrical conductivity of carbon nanotube-based composites", *Applied Physics Letter* 91, 223114 (2007).

P2. W. Lu, T-W Chou and E. Thostenson, "A three-dimensional model of electrical percolation thresholds in carbon nanotube-based composites. *Applied Physics Letter* 96, 223106 (2010).

P3. A. C. Ferrari, J. C. Meyer, V. Scardaci, C. Casiraghi, M. Lazzeri, F. Mauri, S. Piscanec, D. Jiang, K. S. Novoselov, S. Roth and A. K. Geim, "Raman Spectrum of Graphene and Graphene Layers", *Phys. Rev. Lett.*, 2006, 97, 187401–187404.

RESPONSE LETTER

Re. Response to reviewer comments on NCOMMS-18-12592B: "A unified modeling framework for explaining the electrical resistivity trend of polymer composites having segregated structures".

We deeply appreciate the comments of reviewers and have addressed them in the revised manuscript. We thoroughly considered reviewers comments and examined the previously reported papers, which were indicated and thereby, we respectfully explain the originality and importance of our work by highlighting the revised contents in red in the revised manuscript.

The detailed response to the reviewers' comments as follows:

Reviewer #2: *Authors replied to review's comment properly, and the manuscript is also revised and possesses more clear discussion.*

Therefore I can recommend to publish this manuscript to Nature Communications.

RESPONSE:

Authors appreciate careful consideration and efforts of the reviewer.

Reviewer #3: *The paper deals with a numerical investigation supported by experimental results concerning the size effect of the secondary particulate fillers on the electrical conductivity of CNT polymer composites.*

The paper has been brought to my attention after a first review run. The authors have provided suitable responses to the previously issued reviewers' remarks. I agree with the other reviewers that the paper is a valuable contribution to the scientific community, but also after the first review, the paper needs further improvements to be acceptable for publication on a high-qualified journal as Nature Communications.

With respect to the present version of the paper the following further issues should be discussed by the authors.

COMMENT:1. The paper would benefit of a more detailed discussion on which are the novelties with respect to those available in the literature concerning the simulation of electrical conductivity of composites loaded with two types of fillers.

RESPONSE:

As suggested by the reviewer, we have tried to clarify the novelty of the proposed computational modeling approach by stating its principle and advantages. The proposed approach introduce Swiss cheese model that can address the topology of secondary fillers in the composite system. By doing so, a universal computational model that describes the opposite electrical behaviors of the composite system has been established. The following paragraph has been added in the revised manuscript as:

“Conventional approaches with the excluded volume concept over fiber random networks, which capture only monotonicity of the composite percolation, have a limitation in explaining the behavior transitions in electric conductivity of the composite systems. To this end, Swiss cheese model, which enables to account for the distribution and dimension of secondary fillers, is introduced to overcome the failure in the description of the composite system behaviors by the excluded volume.”

COMMENT:2. In the abstract (line 1 and ff.) the authors state “Hybrid carbon nanotube (CNT) composites comprised of more than two different types of fillers...”. The statement is quite misleading since the paper deals with composites with only two types of fillers.

RESPONSE:

We agree with the reviewer’s comment. To clarify the contribution of this work, we have corrected the expression “more than two different types of fillers” to “two different types of fillers. For the future work, we are considering the development of experiments and simulations for the system with more than two different types of fillers.

COMMENT:3. As it concerns the percolation curve (line 201 and ff.) the authors state that the composites follow a power-law characteristic and provide the values of the critical

exponent β . However the reliability of the calculations is not proved, for example by providing the values of the correlation coefficient R^2 for the considered fits.

RESPONSE:

As indicated by the reviewer, we have added further information regarding the critical exponent β and correlation coefficient R^2 as illustrated in plots below. In Figure S8(a), (b) and (c), the DC electrical conductivities (σ_{DC}) of the CNT composites with nano-silica, without silica, and with micro-silica, respectively, follow a power law characteristic of percolation-like behavior ($\sigma_{DC} \sim \sigma_0(p-p_c)^\beta$). The insets also show log–log plots of σ versus $(p - p_c)/p_c$ for the corresponding cases. From the observation on the inset figures, critical exponent (β) and correlation coefficient (R^2) are calculated ((a) composite with nano-silica is 2.7 and 0.98, (b) composite without silica is 1.6 and 0.97, and (c) composite with micro-silica is 1.2 and 0.95). All plots have been added to the supplementary information. In addition, the corresponding contents have been added in the revised manuscript as:

“From the best fit of the σ_{DC} data to log-log plots of the power law, β was evaluated (the nano-silica composites of $\beta \sim 2.7$, the CNT composites without silica of $\beta \sim 1.6$ and the micro-silica composites of $\beta \sim 1.2$ were obtained, as shown in Figure S8).”

(a)

(b)

(c)

Figure S8 The DC electrical conductivity (σ_{DC}) of the CNT composites follows a power law characteristic of percolation-like behavior ($\sigma_{DC} \sim \sigma_0(p-p_c)^\beta$). The inset shows a log–log plot of σ versus $(p - p_c)/p_c$. From the inset figures, critical exponent (β) was evaluated ((a) composite with nano-silica is 2.7, (b) composite without silica is 1.6 and (c) composite with micro-silica is 1.2).

COMMENT:4. With regard to the CNT network the authors state (line 238 and ff.) “The calculation results show that the average number of interconnecting junctions on a single CNT decreases from 23.61 ± 0.74 to 22.58 ± 0.73 . These results correspond to the resistivity increase of the CNT/nano-size silica composite measured in experiment.” The two indicated figures of interconnecting junctions, indeed, are not that different (the lower and upper values of the two ranges are overlapping) to justify the increase of the resistivity values.

RESPONSE:

According to in-depth analyses of the simulation data, we have changed the representative parameter that presents the topological change of the network from “*the average number of interconnecting junctions on a single CNT*” to “*total number of interconnecting junction of wires acting effectively as a conducting path in the percolating clusters that span across the*”

sheet". The new parameter can reflect the contribution of the connecting junctions in active percolating clusters and the number of wires participating in conducting current path simultaneously in characterizing the relationship between the network topology and conductivity. We have recalculated the quantities and updated in the revised manuscript as:

"The topological change of the network leads to the enhancement in the probability of CNTs being interconnected. The total number of interconnecting junction of wires acting effectively as a conducting path in the percolating clusters that span across the 3D simulation domain increases from 502 (± 233) to 2599 (± 438) after incorporating micro-size silica particles. The increase in the total number of interconnecting junctions in the percolating network is an explicit evidence of the resistivity reduction of the network."

and

"The modification of the network topology leads to the decreasing probability of CNTs forming a conducting network. The calculation results show that the total number of interconnecting junctions in active percolating clusters decreases from 502 (± 233) to 269 (± 81). These results correspond to the resistivity increase of the CNT/nano-size silica composite measured in experiment."

COMMENT:5. The authors introduce (line 247 and ff) the "normalized resistance change ... defined as R/R_0 " in order to compare the average resistances of the composite systems with and without silica fillers. It is important to provide the reference value R_0 obtained for the resistance.

RESPONSE:

As suggested by the reviewer, we provide the reference value of R_0 in the manuscript as:

"**Figure 6 (a)** illustrates the resistance change of the CNT network resulting from the modification of the network topology with 1 wt% of 5 μm CNTs. The normalized resistance change is defined as R/R_0 to compare the average resistances of the composite system with and without silica fillers. In this simulation, R_0 corresponds to $3.6 \times 10^4 \Omega$."

and

“Furthermore, **Figure 6 (c)** depicts how the resistance changes as the CNT content increases for various configurations of silica diameters with the CNT length fixed. Note that R_0 changes with respect to the content of CNT, and their values are $3.6 \times 10^4 \Omega$, $7.8 \times 10^2 \Omega$, $1.8 \times 10^2 \Omega$, and $7.1 \times 10^1 \Omega$, for 1-4 wt%, respectively.”

COMMENT:6. Since the main contribution of the paper is on simulation, it should provide more details on the relevant aspects concerning such aspect. In particular, information on geometrical dimensions of the considered domain have to be indicated. In fact (line 220 and ff.), the author state that “Figure 5 (a) shows a random network of CNTs with the diameter of 15 nm and the length of 4 μm leading to the resistance of 193 Ω , which is comparable to the resistance value with the experimental results shown in Figure 4”. However, it is not clear if the dimensions of the simulated domain are the equal to those adopted to carry out the measurements in experimental tests.

RESPONSE:

As indicated by the reviewer, we have revised the following paragraph to clarify the simulation setup as:

“**Figure 5 (a)** shows a random network of CNTs with the diameter of 15 nm and the length of 5 μm in the simulation domain of $10 \mu\text{m} \times 10 \mu\text{m} \times 10 \mu\text{m}$. In the simulation setup, the CNT content of 1 wt% dispersed in the epoxy resin leads to the resistance of $3.6 \times 10^4 \Omega$, which is comparable to the resistivity value of $0.36 \Omega \cdot \text{m}$.”

COMMENT:7. As it concerns the conduction mechanism, the authors state (line 413 and ff.): “...If at least one value of the distances is less than 6nm, which is based on a tunneling mechanism (see section S8 in the Supplementary Information), the two wires are considered to be in contact at the position of the corresponding line segments.”

In Sect. 8 "The maximum value of the tunneling distance d is assumed to be 6 nm implying that two CNTs closer than 6 nm are assumed to be electrically connected." The direct experience of this reviewer which is in accordance with other studies published in literature (see for example P1-P2 in the sequel) is that for a tunneling distance greater than 2nm and height of potential barrier (λ) set at 0.5 eV, the junction is equivalent to an open circuit

(the tunneling resistance is of the order of 10^{12} Ohm). In light of this strong discrepancy with the presented results, the authors should support their indications by a more detailed comparative analysis.

RESPONSE:

We have found the reviewer's comments valid. According to the information provided by the reviewer, the Monte-Carlo simulation has been renovated by setting the maximum value of the tunneling distance to 2 nm, and accordingly, **Figure 6(a)-(c)** have been revised as in the following plots. We see that the revised results obtained from the new simulation still show consistency with the experimental data in **Figure 4**.

Fig. 6 (a) Simulated normalized resistance change (R/R_0) as a function of silica wt% for different silica size. (b) R/R_0 as a function of silica wt% for different CNT lengths. (c) R/R_0 as a function of CNT wt% for different silica size with the CNT length and silica content fixed. Horizontal line indicates CNT composite without silica particle.

COMMENT:8. With respect to the interpretation of the resistivity of the composite (line 303 and ff.) the authors state that “The total resistance of the composite system can be determined by considering two factors: the numbers of the current paths and the average resistivity of the paths... Thus, the total resistance of the composite system decreases as the number of current paths increases, while the total resistance increases as the average resistivity of the paths increase”. Such conclusions however are only qualitatively supported with Fig. 7a and b. From such figures it is quite difficult to estimate what the authors assert “... when the number and resistivity of the paths are small as shown in Figure 7 (a), the change in the number of the current paths mainly affects the change in the total resistance...” . Numerical data would help the reader to understand the authors’ conclusions.

RESPONSE:

We have presented the numerical results on the quantitative characterization based on Voronoi geometry, and have added the detailed explanation on the analysis and the corresponding plots in **Figure 7 (c)** as:

“The impact of the number of current paths and the average resistivity of the paths on the total resistance of the composite system can be characterized quantitatively as illustrated in **Figure 7 (c)**. The number of current paths has a proportional relationship with the average number of CNTs that cross a neck formed between two neighboring fillers. To see this more carefully, we consider the number of silica fillers in the composite system. As the number of the fillers increases, the number of necks subsequently increases. For a fixed number of CNTs in the composite system, only a subset of necks have CNTs crossing them, and only sides of Voronoi tessellations associated with those necks can contribute to the current paths of the percolating network. Thus, the average number of CNT crosses per neck can be used for examining the number of current paths. On the other hand, the average resistivity of a current path can be reinterpreted as the density of CNTs along the sides of Voronoi tessellations since a larger number of CNTs aligned along the side of a Voronoi tessellation can act as a more efficient conducting path. Furthermore, the density of CNTs along the side can be quantified by the density of CNT crosses the neck associated with the side, which is evaluated by the number of CNT crosses divided by the length of the neck. To address both quantitative relationships simultaneously, we introduce the ratio of the average number of CNT crosses per neck to the average neck length as a new *figure of merit* for the total

resistance of the composite system, and numerical results for various configurations are plotted in **Figure 7 (c)**. For four different silica content configurations of 10-40 wt%, the values of the performance metric are compared for micro-silica and nano-silica cases. For clarity, the results for nano-silica fillers are boosted by 150 times since their values are too small compared to the corresponding counter parts with micro-silica fillers. A consistent trend that the metric value of micro-silica fillers exceeds that of nano-silica fillers for a given CNT content indicates that the micro-silica composite system has a higher conductance than the nano-silica composite system. In addition, the contrasting trends of the performance metrics succinctly represents the opposite behaviors of the network resistance for two contrasting filler configurations. These results shed a light on a single parameter characterization of generalized composite systems.”

(c)

Fig. 7 (c) Average number of CNT crosses per neck / average neck length for micro- and nano-silica composites with respect to silica contents.

COMMENT:9. As it concerns the Raman Spectra introduced in Supplementary Information as Fig. S3 the authors state: "Raman spectra illustrated in Figure S3 also confirm the effect of structural changes and the defect of CNTs on the resistivity of the composites. There were no particular changes in the D-band and G-band peaks for all samples, indicating that only reversible deformation of CNTs occurred in the CNT/Silica/PDMS composites without atomic destruction." However, as discussed in the paper P3. cited in the sequel, order and defect level of carbon-based fillers can be identified by the ratio I_D/I_G rather on the observations of the two separate spectra. For CNT 1%/40% micro silica some changes are appreciable.

RESPONSE:

According to the reviewer's comments, we have evaluated the value of the ratio I_D/I_G for CNT composites. The values of I_D/I_G are 0.92, 0.93, and 0.94, for cases of CNT only, micro silica, and nano silica, respectively (see below figure). The deviation of raw CNTs is around 0.1. Thus, no particular changes in the D-band and G-band peaks have been observed for all samples, indicating only reversible deformation of CNTs in the CNT/Silica/PDMS composites without atomic destruction. The plot for the values of I_D/I_G has been added in the supplementary information.

(a)

(b)

Figure S3 (a) Raman spectra of the CNT/Silica/PDMS composites at a wavelength of 514.5 nm to determine changes in the D band (defect and disorder) and the G band (carbon structure). CNT 1 wt% only, CNT 1wt% with micro silica (40 wt%) and CNT 1 wt% with nano silica (30 wt%) are shown. (b) Raman spectra with full range (0 to 3000 cm^{-1})

COMMENT:10. The paper still needs improvements as it concerns English; moreover there are many typos.

RESPONSE:

As suggested by the reviewer, authors have had another round of proofreading of the revised manuscript to improve English use and to correct typos.

Best Regards,

Corresponding author

REVIEWERS' COMMENTS:

Please note that the page and line numbers cited by Reviewer #4 refer to those in the NCOMMS-18-12592A version of the manuscript.

Reviewer #3 (Remarks to the Author):

The authors have thoroughly and accurately revised the manuscript providing a clear answer to this reviewer's questions and comments.

Therefore, the paper can be published on Nature Communications.

I would recommend the authors to provide an additional small correction to the following statement of the revised version of the paper.

"Figure 5 (a) shows a random network of CNTs with a diameter of 15 nm and the length of 5 μm in the simulation domain of 10 μm \times 10 μm \times 10 μm . In the simulation setup, the CNT content of 1 wt% dispersed in the epoxy resin leads to the resistance of $3.6 \times 10^4 \Omega$, which is comparable to the resistivity value of $0.36 \Omega \cdot \text{m}$."

Since resistivity and resistance have not equal dimensions, I would suggest using the term "consistent" rather "comparable".

Reviewer #4 (Remarks to the Author):

The basic idea to improve conductivity in CNT composites is to add filler and reduce the occupied volume of the CNTs thereby increasing the effective volume fraction of CNT and the probability for percolation. However, some recent data has observed larger resistivity. The experimental work in this paper finds that micron sized filler particles reduce the resistivity, but small, nanoscale filler particles result in larger resistivity. The experiments suggest that the fabrication of the system is involved in the differing behavior. The milling process not only disperses the filler particles, but also fractures them and, more importantly, alters the CNT structure. It is thus not clear that this result is general, but whether it depends on the fabrication process. In particular, if one could disperse the filler particles without altering the CNT structure, it is unknown whether the size effect would still occur or not. In this aspect the results of this paper are too specific to meet the standards of Nature Communications. My much stronger major concern is the 'simulations', which are don't appear to be simulations, are poorly described and not physically justified.

The authors state they do Monte Carlo simulations. The description of the simulation method only describes placement of the CNT; there is no description of a Monte Carlo simulation. It is not clear what the authors have done. There is no description of a method for determining a statistical set of structures. With respect to the available description of the computational method, the placement of the CNT is a growth process that involves bending the CNT as it encounters a filler particle. No physical justification for this is given. In the initial mixing of the system, such behavior would not exist. It is possible that the milling process would do this as well as other structural changes to the CNT, but then one needs to model the milling process, which the authors do not do (and probably is too difficult). Overall, there is no justification for the computational procedure as described in the paper, and there are grounds for considering the structures incorrect.

The lack of clarity in the simulation section is unfortunately consistent with the poor writing pervasive in the section. Most paragraphs have mistakes some of which make sentences incomprehensible. Here are some specific examples:

Lines 79-81: From this Swiss cheese model, a Voronoi tessellation associated with each of voids can be obtained by collecting the points in the volume closer to the center of a secondary filler than other fillers (Figure 1 (c)).

1) a Voronoi tessellation is not obtained by 'collecting points'; lines perpendicular to the midpoint are drawn.

2) 'secondary filler other than fillers' means what? As written it means an empty set which is problematic!

The next sentence claims Fig. 1c "presents all possible configuration", which does not match singular and plural. In addition, the figure presents only one possible pathway for a conducting path. There can be other adjacent pathways.

lines 95-96: "..., the transition in the excluded-volume effect ..." There is no transition "in" the excluded volume. There could be a transition as a function of excluded volume.

lines 97 references an hypothesis, but the hypothesis has not be specifically stated.

line 111: "To avoid ..." just gives the definition of avoiding overlap. There is no need for this sentence.

"bendable wire" is unclear terminology. Is it a flexible line? cylinder? or what?

line 127: "The length of a line segment is set identical to the width of the wire ..."

The 'wire' has never been defined (and see previous) and thus the reader does not know what its width is.

RESPONSE LETTER

Re. Response to reviewer comments on NCOMMS-18-12592B: "A unified modeling framework for explaining the electrical resistivity trend of polymer composites having segregated structures".

We deeply appreciate the comments of reviewers and have addressed them in the revised manuscript. We thoroughly considered reviewers comments and examined the previously reported papers, which were indicated and thereby, we respectfully explain the originality and importance of our work by highlighting the revised contents in red in the revised manuscript.

The detailed response to the reviewers' comments as follows:

Reviewer #3: *The authors have thoroughly and accurately revised the manuscript providing a clear answer to this reviewer's questions and comments.*

Therefore, the paper can be published on Nature Communications.

I would recommend the authors to provide an additional small correction to the following statement of the revised version of the paper.

"Figure 5 (a) shows a random network of CNTs with a diameter of 15 nm and the length of 5 μm in the simulation domain of $10 \mu\text{m} \times 10 \mu\text{m} \times 10 \mu\text{m}$. In the simulation setup, the CNT content of 1 wt% dispersed in the epoxy resin leads to the resistance of $3.6 \times 10^4 \Omega$, which is comparable to the resistivity value of $0.36 \Omega\cdot\text{m}$."

Since resistivity and resistance have not equal dimensions, I would suggest using the term "consistent" rather "comparable".

RESPONSE:

Authors appreciate careful consideration and efforts of the reviewer. We have changed the corresponding expression as suggested.

Reviewer #4: *The basic idea to improve conductivity in CNT composites is to add filler and reduce the occupied volume of the CNTs thereby increasing the effective volume fraction of CNT and the probability for percolation. However, some recent data has observed larger resistivity. The experimental work in this paper finds that micron sized filler particles reduce*

the resistivity, but small, nanoscale filler particles result in larger resistivity. The experiments suggest that the fabrication of the system is involved in the differing behavior. The milling process not only disperses the filler particles, but also fractures them and, more importantly, alters the CNT structure. It is thus not clear that this result is general, but whether it depends on the fabrication process. In particular, if one could disperse the filler particles without altering the CNT structure, it is unknown whether the size effect would still occur or not. In this aspect the results of this paper are too specific to meet the standards of Nature Communications.

RESPONSE:

Ideal dispersion of fillers in polymer matrix is critical. In particular, poor dispersion conditions of two fillers often cause opposite or random trends in electrical conductivity despite their similar structure and materials. In order to obtain a consistent composite conductivity, both fillers should be dispersed uniformly within the polymer matrix. One way of achieving uniform dispersion is to use the 3-roll milling process. The 3-roll milling process disperses nano/micro materials uniformly in uncured polymer resin (in liquid state) as shown in Figure R1: High shear force of 3 rolls (described in detail in [1]) rolls entangled CNT aggregations and heavy CNT bundles out into a composite spread.

Nanocomposite fabrication (3 roll milling step)

Figure R1. Scheme of CNT dispersion using high shear force of 3 roll milling

[1] Thielemann, P., Yun, D. J., Asbeck, P., Bandaru, P. R., & Park, S. H. Superior electromagnetic interference shielding and dielectric properties of carbon nanotube composite through the use of high aspect ratio CNTs and three-roll milling. *Org. Electron.* **12**, 1531–1537 (2013).

This process may cause some structure changes in CNTs. To see this, we have conducted morphological analysis in shortening effects on CNTs. As-received CNTs range within 10-15 μm long as shown in Supplementary Figure 3 and 4. However, the length of a CNT measured after three-roll milling press reduces to the range of 3-6 μm (mostly measured around 4 μm). Thus, the length of CNT has been specified to 4 μm for computational modeling. Along with the length change in CNT, the decrease in silica particle size was also observed. While the size of as-received micro-silica particles uses to ranges between 7 to 10 μm as shown in Supplementary Figure 5 and 6, the milling process has reduced the micro-silica size to 2-4 μm (mostly measured around 3 μm). From this observation as also shown in Supplementary Figure 5, 3 μm has been considered for the size of micro-silica particles in simulation. Raman spectra illustrated in Supplementary Figure 7 have been conducted to confirm the invariance of carbon structure. There were no particular changes in the D-band and G-band peaks for all tested samples. This reveals that no atomic destruction occurs in CNTs and that the resistivity change is not attributed to such atomic structure change. This discussion has been added in the revised manuscript and supplementary information.

Supplementary Figure 3 Distributions of CNT length before/after three-roll milling

Supplementary Figure 4 SEM images of CNT before/after three-roll milling

Supplementary Figure 5 Distributions of micro-silica particle size before/after three-roll milling

Supplementary Figure 6 SEM images of micro-silica particles before/after three-roll milling

(a)

(b)

Supplementary Figure 7 (a) Raman spectra of the CNT/Silica/PDMS composites at a wavelength of 514.5 nm to determine changes in the D band (defect and disorder) and the G band (carbon structure). CNT 1 wt% only, CNT 1wt% with micro silica (40 wt%) and CNT 1 wt% with nano silica (30 wt%) are shown. (b) Raman spectra with a full range from 0 to 3000 cm^{-1}

My much stronger major concern is the 'simulations', which are don't appear to be simulations, are poorly described and not physically justified.

The authors state they do Monte Carlo simulations. The description of the simulation method only describes placement of the CNT; there is no description of a Monte Carlo simulation. It is not clear what the authors have done. There is no description of a method for determining a statistical set of structures. With respect to the available description of the computational method, the placement of the CNT is a growth process that involves bending the CNT as it encounters a filler particle. No physical justification for this is given. In the initial mixing of the system, such behavior would not exist. It is possible that the milling process would do this as well as other structural changes to the CNT, but then one needs to model the milling process, which the authors do not do (and probably is too difficult). Overall, there is no justification for the computational procedure as described in the paper, and there are grounds for considering the structures incorrect.

RESPONSE:

As suggested by the reviewer, we have added the details of the Monte Carlo simulation including how to define the simulation domain, how to obtain a CNT network, how to calculate the resistance, how to evaluate an average, and the implementation issue, along with an appropriate reference in the revised manuscript as:

“A sample cubic domain of $10\ \mu\text{m} \times 10\ \mu\text{m} \times 10\ \mu\text{m}$ is taken from the composite model to obtain the CNT network. The connectivity of the obtained CNT network is checked by examining the contacts between flexible wires.”

and

“A clustering analysis identifies percolating clusters of wires that span across the domain. Starting from the leftmost wires touching the left end, each cluster is expanded gradually toward the right end by including wires in contact. If a cluster contains one of the rightmost wires touching the right end, this cluster is declared to be a conducting path. For all conducting paths, Kirchhoff's current law (KCL) is applied at all junctions of wires to construct a system of linear equations about the voltage drops across the junctions due to the contact resistivity (ρ_c). In the simulation, ρ_c is considered as a tunneling resistivity arising from the thin insulating layer (i.e., epoxy resin) between crossing CNTs.⁴⁴⁻⁴⁷ Since 1 V voltage is assumed to apply across the domain, the inverse of the solution of the system of linear equations corresponds to the overall resistance of the network. This process is repeated to obtain 500 independent random instances, and an ensemble average of the resistance is

calculated based on the set of tested samples between 30th percentile and 70th percentile. The simulation is conducted over different configurations of silica content, silica size, CNT content, and CNT dimension. For massive simulation, the source code is implemented in MATLAB parallel computing tool box.⁴⁸

48. Parallel Computing Toolbox Release 2017b. The MathWorks, Inc., Natick, MA, United States.

Furthermore, we have addressed the justification of the creation of CNT wires in the composite domain. According to the role of the 3-roll milling process, highly loaded nano/micro second filler particles are dispersed uniformly over a polymer matrix. The 3-roll milling process increases the viscosity of CNT/silica/PDMS very high since the dispersion enlarge the contact surface area between the matrix and the second fillers, thereby causing the increase in the shear stress of the mixture. As a result, there are no re-aggregation and re-orientation among CNTs and silica particles in the mixture. The SEM and TEM images in Figure 2 and 3 prove this by showing that CNTs and silica particles were distributed uniformly in the entire polymer matrix after the 3-roll milling process. Thus, the description of the steady state of the mixture suffices to simulate the electrical characteristics of the CNT composites instead of modeling the 3-roll milling process itself in a computational way.

According to morphological characteristics observed in the images, the Monte-Carlo simulation places a random instance of the CNT distribution by considering the following properties only:

1. CNTs and silica particles are distributed uniformly in the simulation domain.
2. CNTs can pass through a gap between silica particles but cannot penetrate the particles themselves.

To model those properties, the position and direction of a flexible CNT wire are chosen randomly by choosing the position and direction of the initial segment at random. A series of line segments proceeds forwardly (which can also be empirically observed, in particular, in Fig 3(b)), and the direction of a line segment in contact with a silica particle changes its direction to circumvent the particle (which can also be empirically observed, in particular, in Fig 3(d) and (f)).

The CNT networks built by the resulting Monte-Carlo simulation strategy simulates CNT bundles concentrated between micro silica particles, while twisted CNTs are found between

nano silica particles, each of the both shapes is observed from the TEM images. In addition, a good agreement of the electrical conducting characteristics calculated from the CNT dispersion with the experimental results shows that the proposed simulation model can properly address the composite systems.

We have added the corresponding comments in Simulation section of the revised manuscript.

Fig. 3 Graphical scheme and TEM image. (a) and (b) CNT composite without silica. (c) and (d) Nano-silica fillers with diameter of 20 nm. (e) and (f) Micro-silica fillers with diameter of 3-4 μm .

The lack of clarity in the simulation section is unfortunately consistent with the poor writing pervasive in the section. Most paragraphs have mistakes some of which make sentences incomprehensible. Here are some specific examples:

Lines 79-81: From this Swiss cheese model, a Voronoi tessellation associated with each of voids can be obtained by collecting the points in the volume closer to the center of a secondary filler than other fillers (Figure 1 (c)).

1) a Voronoi tessellation is not obtained by 'collecting points'; lines perpendicular to the midpoint are drawn.

2) 'secondary filler other than fillers' means what? As written it means an empty set which is problematic!

The next sentence claims Fig. 1c "presents all possible configuration", which does not match singular and plural. In addition, the figure presents only one possible pathway for a conducting path. There can be other adjacent pathways.

RESPONSE:

As suggested by the reviewer, we have clarified the formal definition of a Voronoi tessellation as:

“In this Swiss cheese model, a Voronoi tessellation associated with each of voids is the region consisting of all points closer to the center of the void than to those of any other voids (Figure 1 (c)).”

According to the definition of Swiss cheese model, a side of a Voronoi tessellation can act as a conducting path in the medium between voids, which correspond to silica particles in the CNT composite. The collection of those sides form the set of all connected paths that can possibly exist between silica particles and traverse the CNT composite. Thus, the collection of sides of Voronoi tessellations suffices to characterize conducting paths in the CNT-silica composite configuration, as shown in Fig 1c. In addition, we have corrected the sentences as:

“A network formed with the collected sides of Voronoi tessellations (represented in grey solid lines in Figure 1 (c)) presents the collection of all conducting paths in the polymer matrix with particulate fillers.”

lines 95-96: "..., the transition in the excluded-volume effect ..." There is no transition "in" the excluded volume. There could be a transition as a function of excluded volume.

RESPONSE:

To avoid a confusion about the transition, which is intended to indicate the change in the trend of the electrical conductivity of composite according to the size of the secondary particles, we have modified the sentence as:

“In this study, the transition in the impact on the electrical conductivity is observed in accordance with the excluded volume caused by secondary particles, i.e., the excluded volume improves/prevents the conductivity of CNT composites depending on the particle size.”

lines 97 references an hypothesis, but the hypothesis has not be specifically stated.

RESPONSE:

As suggested by the reviewer, we have specified the hypothesis as:

“This sets up a hypothesis that the network morphology of CNT composites depends on the size of the second fillers, leading to the variation of the electrical conductivity. The hypothesis is proved in experimental and theoretical ways.”

line 111: "To avoid" just gives the definition of avoiding overlap. There is no need for this sentence.

"bendable wire" is unclear terminology. Is it a flexible line? cylinder? or what?

RESPONSE:

As suggested by the reviewer, we have removed the redundancy to change the corresponding sentence. In addition, we have changed a bendable wire into a flexible wire and added the definition of a flexible wire as:

“Note that, in this simulation, a flexible CNT wire in the composites is modeled as a long thin cylindrical object with many joints in order to consider the bending of the wire.

Spherical volumes representing silica particles are scattered in a cubic domain by choosing at random central coordinates of individual particles apart by farther than the diameter.”

line 127: "The length of a line segment is set identical to the width of the wire"

The 'wire' has never been defined (and see previous) and thus the reader does not know what its width is.

RESPONSE:

As suggested by the reviewer, we have added the definition of wire as in the response to the previous comment. Also, we have added a statement that the width of a single wire is obtained from the average diameter of CNTs used in the experiment as:

“In this simulation, the width and length of a single wire are obtained from the average diameter and length of CNTs used in the experiment, respectively. The length of a line segment contained in a flexible wire is set identical to the width of the wire so that the contact of two flexible wires can be examined simply (See Supplementary Figure 1 and 2).”

EDITORIAL REQUESTS:

TITLE PAGE

** For clarity and concision, I suggest the following revision I suggest the following revision to the title:*

'Modeling the electrical resistivity of polymer composites with segregated structures'

If you would like to suggest an alternative title, please ensure that it does not exceed 15 words and does not contain punctuation.

RESPONSE:

We have changed the title as suggested.

** Please shorten the abstract to 150 words or fewer.*

RESPONSE:

We have shortened the abstract to the range within 150 words.

MAIN TEXT

** The main text should include only the following sections: Introduction, Results, and optional Discussion, each of which must begin with a heading. All other section headings should be removed or renamed.*

RESPONSE:

We have removed additional headings except Introduction, Results, and Discussion.

** Please rearrange the Introduction so that all discussion of previous work appears first. The final paragraph should contain only a concise summary of the current work, in the present tense.*

RESPONSE:

We have tried to revise Introduction as indicated.

LANGUAGE AND STYLE

** Please do not use italics or bold font to convey emphasis (in both the main text and the display items).*

RESPONSE:

We have removed all italics and bolds in the manuscript.

** Please make sure that mathematical terms throughout your manuscript and Supplementary Information (including in figures, figure axes, and legends) conform strictly to the following guidelines. Equations should be supplied in editable format, and not as images. Scalar variables (e.g. x , V , χ) should be typeset in italic, whereas multi-letter variables should be formatted in roman. Constants (e.g. \hbar , G , c) should be typeset in italics (the only exceptions being e , i , π , which should be typeset in Roman) and vectors (such as r , the wavevector k , or the magnetic field vector B) should be typeset in bold without italics. In contrast, subscripts and superscripts should only be italicised if they too are variables or constants. Those that are labels (such as the 'c' in the critical temperature, T_c , the 'F' in the Fermi energy, E_F , or the 'crit' in the critical current, I_{crit}) should be typeset in roman. To avoid doubt, unit dimensions should be expressed using negative integers (e.g. $\text{kg m}^{-1} \text{s}^{-2}$, not kg/ms^2) or the word 'per'.*

RESPONSE:

We have corrected all variables and constants as indicated.

** Please double-check the first sentence of the conclusions.*

RESPONSE:

We have corrected the corresponding sentence as suggested by the comment on end note.

METHODS AND DATA

** All Nature Communications manuscripts must include a section titled "Data Availability" as a separate section after the Methods section and before the References. For more information on this policy, and a list of examples, please see*

<http://www.nature.com/authors/policies/data/data-availability-statements-data-citations.pdf>

RESPONSE:

We have added the section on code availability.

** DATA SOURCES: We strongly encourage authors to deposit all new data associated with the paper in a persistent repository where they can be freely and enduringly accessed. We recommend submitting the data to discipline-specific, community-recognized repositories, where possible and a list of recommended repositories is provided here:*

<http://www.nature.com/sdata/policies/repositories>

RESPONSE:

We have added the section on data availability.

** For all studies developing new software or using custom code that is deemed central to the conclusions, a statement must be included, under the subheading "Code Availability", indicating whether and how the code can be accessed, including any restrictions to access. This section should also include information on the versions of any software used, if relevant, and any specific variables or parameters used to generate, test, or process the current dataset. Code availability statements should be provided as a separate section after the Data Availability section but before the References. Please see our policy on code availability for more information. <http://www.nature.com/sdata/for-authors/editorial-and-publishing-policies#code-avail>*

RESPONSE:

We have added the section on code availability after data availability.

END NOTES

** Please edit the sentence*

'In summary, we have shown that the conductivity variation of CNT composite having secondary fillers depend on the filler size which has confirmed through the combination of Voronoi geometry and percolation theory.'

to

'In summary, we have shown that the conductivity variation of CNT composites containing secondary fillers depends on the filler size, which has been confirmed through the combination of Voronoi geometry and percolation theory.'

for clarity.

RESPONSE:

We have changed the sentence as suggested.

DISPLAY ITEMS

** Please check whether your manuscript or Supplementary Information contain third-party images, such as figures from the literature, stock photos, clip art or commercial satellite and map data. We strongly discourage the use or adaptation of previously published images, but if this is unavoidable, please request the necessary rights documentation to re-use such material from the relevant copyright holders and return this to us when you submit your revised manuscript.*

RESPONSE:

We have confirmed that no third-party materials are contained in the manuscript.

** Please include a brief title for all figure legends that summarise the whole figure and does not refer to specific panels.*

RESPONSE:

We have changed all captions to summarize the findings represented by figures.

** Please ensure that each display item is no larger than a single A4 portrait page (260x179 mm).*

RESPONSE:

We have confirmed that all graphics fit within a page.

** Please define any new abbreviations, symbols or colours present in your figures in the associated legends. Please do not use symbols in your legend, instead please write out the symbols in words (blue circles, red dashed line, etc.).*

RESPONSE:

We have removed symbols in figure captions and written out them in words.

** In each figure and supplementary figure where error bars are used, they must be defined. One statement at the end of each figure is sufficient if the error bars are equivalent throughout the figure.*

Please define the error bars displayed in Fig.4.

RESPONSE:

We have added the definition of error bars in Figure 4.

SUPPLEMENTARY INFORMATION

** We do not edit Supplementary Information files; they will be uploaded with the published article as they are submitted with the final version of your manuscript. Any tracked changes should be removed from the file.*

RESPONSE:

We have confirmed that no tracked change exists in the Supplementary Information.

** In the Supplementary Information file and the main manuscript text, supplementary items must be labelled and cited using only the following formats: Supplementary Figure 1, Supplementary Table 1, Supplementary Methods, Supplementary Note 1, Supplementary Discussion, and Supplementary References. Please note the use of "Supplementary" and that we do not use the "S" prefix.*

RESPONSE:

We have corrected the citation to Supplementary Information in the manuscript and reorganized the labels as indicated.

** Please replace general citations to the Supplementary Information (e.g. "see Supplementary Information") with specific citations (e.g. "See Supplementary Figure 1", etc.).*

RESPONSE:

We have corrected the citation to Supplementary Information in the manuscript and reorganized the labels as indicated.

** Please label supplementary equations sequentially as (1), (2), (3), etc. (and without an "S" prefix).*

RESPONSE:

We have reorganized the labels of equations in the Supplementary Information by removing the prefix.

** Supplementary References should appear at the end of the Supplementary Information file, and should be self-contained and numbered from 1. References mentioned in both the main text and the Supplementary Information should be part of both reference lists so that the*

Supplementary Information does not refer to the reference list in the main paper and vice versa.

RESPONSE:

We have confirmed that supplementary references are self-contained.

** Your paper will be accompanied by a two-sentence Editor's summary, of between 250-300 characters including spaces, when it is published on our homepage. Could you please approve the draft summary below or provide us with a suitably edited version.*

"Carbon nanotube-polymer composites containing secondary fillers are thought to possess enhanced electrical and mechanical properties. Here the authors combine Monte Carlo calculations with resistivity experiments to study the effect of filler size and shape on electrical conductivity."

RESPONSE:

We approve the draft summary provided above.

Best Regards,

Corresponding author